# Fairness and Disentanglement: A Critical Review of Predominant Bias in Neural Networks

**Jiazhi Li**                                                                    *jiazhil@usc.edu*
*Ming Hsieh Department of Electrical and Computer Engineering*
*University of Southern California*

**Mahyar Khayatkhoei**                                                           *mkhayat@isi.edu*
*Information Sciences Institute*
*University of Southern California*

**Jiageng Zhu**                                                                  *jiagengz@usc.edu*
*Ming Hsieh Department of Electrical and Computer Engineering*
*University of Southern California*

**Hanchen Xie**                                                                  *hanchenx@usc.edu*
*Thomas Lord Department of Computer Science*
*University of Southern California*

**Mohamed E. Hussein**                                                           *mehussein@apple.com*
*Apple Inc.*

**Wael AbdAlmageed**                                                             *wabdalm@clemson.edu*
*Holcombe Department of Electrical and Computer Engineering*
*Clemson University*

**Reviewed on OpenReview:** *https://openreview.net/forum?id=LLiJ1WsL2e&referrer=%5BAuthor% 20Console%5D(%2Fgroup%3Fid%3DTMLR%2FAuthors%23your-submissions)*

## Abstract

Bias issues of neural networks garner significant attention along with their promising advancement. Among various bias issues, mitigating two predominant biases is crucial in advancing fair and trustworthy AI: (1) ensuring neural networks yield even performance across demographic groups, and (2) ensuring algorithmic decision-making does not rely on protected attributes. However, upon the investigation of 415 papers in the relevant literature, we find that there exists a persistent, extensive but under-explored confusion regarding these two types of biases. Furthermore, the confusion has already significantly hampered the clarity of the community and the subsequent development of debiasing methodologies. Thus, in this work, we aim to restore clarity by providing two mathematical definitions for these two predominant biases and leveraging these definitions to unify a comprehensive list of papers. Next, we highlight the common phenomena and the possible reasons for the existing confusion. To alleviate the confusion, we provide extensive experiments on synthetic, census, and image datasets to validate the distinct nature of these biases, distinguish their different real-world manifestations, and evaluate the effectiveness of a comprehensive list of bias assessment metrics in assessing the mitigation of these biases. Further, we compare

these two types of biases from multiple dimensions, including the underlying causes, debiasing methods, evaluation protocol, prevalent datasets, and future directions. Last, we provide several suggestions aiming to guide researchers engaged in bias-related work to avoid confusion and further enhance clarity in the community.

# 1 Introduction

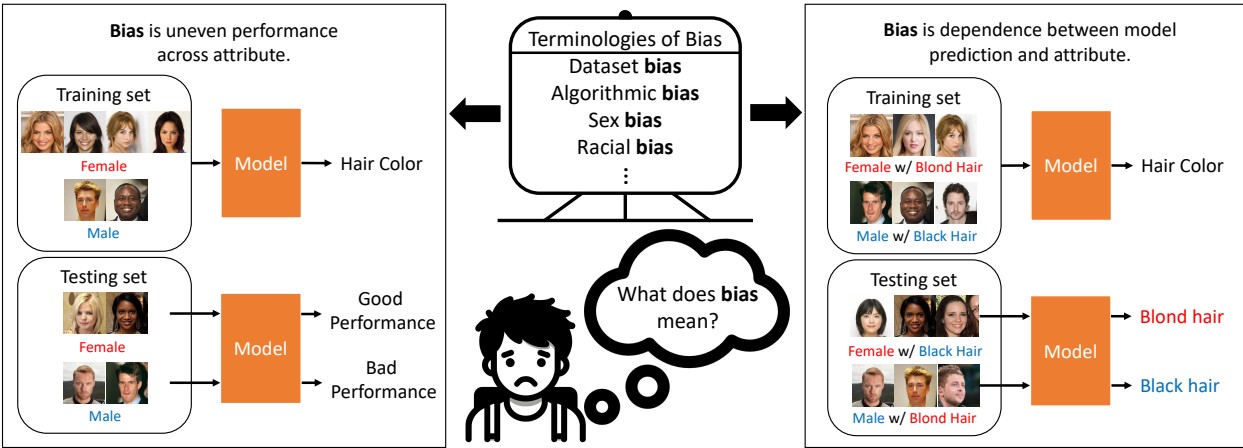

Figure 1: The same set of terminology about bias is interpreted differently by experts, which significantly confuses the understanding of the audience. By investigating 415 papers about prevalent bias issues, we discover that there exists significant confusion regarding these prevalent bias issues. The confusion is evident in several ways, such as ambiguity of terminology, inaccurate motivation, and lack of terminology reuse. Most notably, several studies inaccurately motivate themselves on a particular bias while actually addressing a different type of bias. This prevailing confusion considerably impedes the clarity of related work. Thus, we propose new definitions to unify the existing literature and pave a clear path for future research.

Neural networks have shown promising advances in many prediction and classification tasks (Russakovsky et al., 2015; He et al., 2016; Mnih et al., 2013). Along with the impressive capability of neural networks, its societal impact has garnered great attention (Buolamwini & Gebru, 2018; Gong et al., 2021), particularly regarding *protected attributes* (*e.g.*, sex, race, and age), which cannot be used in the decision-making process (Corbett-Davies & Goel, 2018). Failing to carefully consider protected attributes while deploying neural networks can lead to bias issues and severely compromise fairness for specific demographic groups in various real-world applications (Angwin et al., 2022a; Li & AbdAlmageed, 2024). For instance, facial recognition systems may more correctly recognize males than females (Gong et al., 2020). Besides, Artificial Intelligence-assisted bank loan systems may classify a higher proportion of male applicants as having bad credit than female applicants (Zhu et al., 2021).

The underlying bias issues of neural networks, involved in the aforementioned examples, lead to important discussions (Kim et al., 2019a; Liu et al., 2022a; Tartaglione et al., 2021). Specifically, these aforementioned examples highlight the presence of two distinct prevalent types of biases. Without loss of generality, for disambiguation, these two predominant biases can be summarized as follows:

Table 1: Main distinctions between Type I Bias and Type II Bias.

|  | Type I Bias | Type II Bias |
| --- | --- | --- |
| Manifestation | Uneven performance across attributes | Dependence between model prediction and attribute |
| Use of ground truth | ✓ | ✗ |
| Representative example | Facial recognition systems exhibit lower performance in one demographic group compared to others | Bank loan systems tend to approve loans more frequently for one demographic group compared to others |
| Possible reason | Insufficient training in the underrepresented group | Correlation between the target $Y$ and the attribute in the training set |

- The model yields uneven performance across different demographic attributes, referred to as *Type I Bias.*

- The model depends on demographic attributes to make predictions, referred to as *Type II Bias.*

Although these two prevalent types of biases differ in many aspects, as highlighted in Tab. 1, the current literature often ambiguously groups them under the general term "bias" (*e.g.*, dataset bias, algorithmic bias, sex bias, or racial bias) (Alvi et al., 2018; Ragonesi et al., 2021) and interpret them differently across scenarios. Furthermore, numerous works addressing one type of bias inadvertently cite the other as their motivation (Wang et al., 2019b; Wang & Deng, 2020; Salvador et al., 2022). Additionally, existing survey papers may lack a clear taxonomy that sufficiently distinguishes between them or explicitly acknowledge their fundamental distinctions (Mehrabi et al., 2021a; Wang et al., 2022c; Castelnovo et al., 2022).

Overlooking the distinction between these two types of biases significantly compromises clarity in the current literature and leads to various negative consequences. Specifically, for new researchers, the lingering question of which specific type of bias a paper addresses creates unnecessary confusion. As illustrated in Fig. 1, this confusion arises when researchers encounter multiple works that all reference "bias" without clearly specifying its nature. The widespread confusion surrounding these biases and the lack of clear definitions to separate them results in weak motivation, ambiguous statements, and vague contributions in the existing debiasing work, significantly impeding the clarity of the associated research. Additionally, persistent conflation of these biases, usage of inappropriate references, and unfair comparison between methods addressing different biases can lead to an expanding misunderstanding over time. Besides, this confusion complicates the resolution of bias issues and hinders the advancement of future work in this field.

To that end, the main goal of this paper is to unify the existing literature about Type I Bias and Type II Bias, rectify the common confusion regarding them, and alleviate the cognitive burden for future research. The contributions of this paper can be summarized as follows:

- Proposing General mathematical definitions for Type I Bias and Type II Bias (Sec. 2) and providing a summary of their corresponding related work (Sec. 7). These can be utilized as a roadmap for future work.

- Unifying a comprehensive list of work and relevant fairness criteria under the definition of Type I Bias and Type II Bias (Sec. 4).

- Elucidating the existing phenomena stemming from the confusion between Type I Bias and Type II Bias (Sec. 5.1), and exploring the underlying reasons that contribute to the confusion (Sec. 5.2).

- Conducting extensive experiments to examine the distinction between Type I Bias and Type II Bias (Sec. 6).

- Offering some suggestions to foster a clear community regarding these bias issues (Sec. 8).

## 2 Definitions

To define and distinguish these two types of biases, we first establish several key concepts. Let the dataset $\mathcal{D}$ be a set of instances $(x, y, z)$ where each sample $x \in \mathcal{X}$ is annotated with a ground truth label $y \in \mathcal{Y}$ for a downstream task (*e.g.*, identity in face recognition) and an attribute label $a \in \mathcal{A}$ (*e.g.*, sex). A model $f : \mathcal{X} \to \mathcal{Y}$ maps an input $x$ to a predicted label $\hat{y}$. In this section, we introduce formal mathematical definitions for these two types of biases, referred to as Type I Bias and Type II Bias, which will be consistently used throughout the paper. In the following sections, we will review 415 papers to demonstrate that various commonly discussed bias issues can be unified using these definitions and explore the phenomena and reasons behind the existing confusion between these bias issues. Furthermore, in Sec. 6, we present a case study using both synthetic and real-world datasets to demonstrate scenarios in which Type I Bias occurs without Type II Bias, and vice versa.

### 2.1 Type I Bias

The manifestation of Type I Bias is uneven model performance across different demographic groups (Wang et al., 2019b; Gong et al., 2021; Liu et al., 2022a). Specifically, model performance can be evaluated using various metrics, *e.g.*, error rate (Buolamwini & Gebru, 2018), loss (Hashimoto et al., 2018), accuracy (Kim et al., 2019b), average precision (AP) (Ramaswamy et al., 2021), positive predictive value (PPV), true positive rate (TPR) (Dhar et al., 2021; Adeli et al., 2021), false positive rate (FPR) (Xu et al., 2021c), average false rate (AFR), mean AFR (M AFR) (Ryu et al., 2017), confusion matrix (Gong et al., 2020), F1 score (Adeli et al., 2021), receiver operating characteristic curve (ROC) (Wang & Deng, 2020), area under the ROC (AUC) (Mirjalili et al., 2019). All these metrics can be unified under the format of a distance measure $d(\hat{Y}, Y)$, evaluated based on model prediction $\hat{Y}$ and ground truth label $Y$. Thus, we can formally define this type of bias as follows:

**Definition 1.** *Type I Bias. A model $f$ involves* Type I Bias *if $f$ yields uneven performance $d(\hat{Y}, Y)$ across attribute $A$,*

$$\sup_{a,a'\in\mathcal{A},d\in\mathcal{M}} |d(\hat{Y}, Y|A = a) - d(\hat{Y}, Y|A = a')| > 0 \tag{1}$$

where $a, a'$ are possible values of $A$ (*e.g.*, female and male), and $\mathcal{M}$ is the set of all potential performance metrics.

### 2.2 Type II Bias

On the other hand, the manifestation of Type II Bias is dependence between model prediction and attribute (Alvi et al., 2018; Kim et al., 2019a; Wang et al., 2020b; Nam et al., 2020). Specifically, these attributes can be categorized by sensitive/protected attributes (Lokhande et al., 2020) (*e.g.*, sex in creditworthiness prediction) or spurious attributes (Sagawa* et al., 2020) (*e.g.*, texture in object recognition). Both of these scenarios can be unified as the dependence between model prediction and the specific attribute. Thus, we can formally define this type of bias as follows:

**Definition 2.** *Type II Bias. A model $f$ involves* Type II Bias *if model prediction $\hat{Y}$ is not independent with attribute $A$,*

$$\sup_{a,a'\in\mathcal{A}} |P(\hat{Y}|A = a) - P(\hat{Y}|A = a')| > 0 \tag{2}$$

where $a, a'$ are possible values of $A$ (*e.g.*, female and male).

## 3 Method

In this section, we introduce the method used to conduct the investigation on a set of 415 papers that discuss relevant bias issues. Specifically, to construct the initial set of relevant work, we search for the keywords "bias" or "fair" in the title of papers from NeurIPS, ICML, ICLR, and FAccT published before February 2025. We include papers that discuss bias issues whose manifestation aligns with either Type I Bias or Type II Bias (we will detail the unification in Sec. 4). We exclude papers that address other bias issues such as inductive bias (Baxter, 2000; Zietlow et al., 2021), implicit bias (FitzGerald & Hurst, 2017; Camuto et al., 2021), selection bias (Hernán et al., 2004; Akbari et al., 2021), sampling bias (Winship & Mare, 1992; Xu et al., 2022a), spectral bias (Fang & Xu, 2024), exposure bias (Li et al., 2024) or bias-variance (Ha et al., 2024; Chen et al., 2024b). Furthermore, to ensure we do not overlook any relevant papers without these keywords or from other prominent conferences such as CVPR, ICCV, and ECCV, we manually traversal the citation graph of the paper in the initial set and append the relevant papers that are either cited by or cite the papers in the initial set.

Once we identify the scope of the investigated papers, we read these papers to determine which type of bias they address by examining two aspects: the problem statement and evaluation protocol. We will elaborate on the criterion for categorizing papers into our definitions in Sec. 4. To accommodate the recent emerging

direction of addressing unlabeled and unknown bias, we enrich the taxonomy with an additional dimension about the status of attribute $A$. As shown in Tab. 2, we count the number of papers in each category. Note that the total number is not equal to 415 since some papers address both types of biases. We present the categorization list of all 415 investigated papers in the Appendix.

Table 2: The taxonomy of bias issues based on 415 papers.

| Type of Bias | Attribute $A$ | | Papers | Examples |
|---|---|---|---|---|
| | Known | Labeled | | |
| Type I Bias | ✓ | ✓ | 253 | Gong et al. (2020; 2021); Wang & Deng (2020) |
| | ✓ | ✗ | - | - |
| | ✗ | ✗ | - | - |
| Type II Bias | ✓ | ✓ | 246 | Kim et al. (2019a); Zhu et al. (2021); Tartaglione et al. (2021) |
| | ✓ | ✗ | 8 | Wang et al. (2019a); Bahng et al. (2020); Cadene et al. (2019) |
| | ✗ | ✗ | 30 | Nam et al. (2020); Zhao et al. (2023a); Jeon et al. (2022) |
| Survey | - | - | 25 | Mehrabi et al. (2021a); Du et al. (2020); Castelnovo et al. (2022) |

## 4    Unification

In this section, we clarify how bias issues discussed in existing literature align with our proposed definitions. Generally, we categorize the bias into a specific type of bias in our definition if the presence of this bias implies the existence of bias in our definition. Furthermore, the categorization primarily relies on two key factors: the manifestation of bias issues explicitly addressed (if stated in "Problem Statement" section) and the characteristics of evaluation protocol[1]. Other aspects, such as motivation, related work, method, or bias assessment, are considered secondary factors for categorization. This is because certain papers, despite addressing different manifestations of bias, can exhibit similarities in these aspects, thereby leading to the confusion between these two types of biases, as elaborated in Sec. 5.1.

### 4.1    Type I Bias

The general form of Type I Bias is characterized by the uneven performance of the target across attributes. This definition can be extended to unify a wide range of papers by specifying the usage of performance metrics and the kind of target. To clarify, several representative descriptions are shown as follows, *e.g.*,

- *"Racial bias indeed degrades the fairness of recognition system and the error rates on non-Caucasians are usually much higher than Caucasians."* (Wang & Deng, 2020)

- *"A certain demographic group can be better recognized than other groups."* (Gong et al., 2021)

- *"Recognition accuracies depend on the demographic cohort."* (Wang et al., 2019b)

By specifying how performance is evaluated, Type I Bias covers a broad range of papers where model performance is evaluated using various criteria such as loss (Hashimoto et al., 2018) and accuracy (Kim et al., 2019b). Furthermore, by specifying the kind of target, this definition can unify a wider range of papers. For instance, considering sex as an attribute, the targets can include identity (Gong et al., 2020; Salvador et al., 2022) (*e.g.*, face recognition), the attribute itself (Buolamwini & Gebru, 2018; Karkkainen & Joo, 2021) (*e.g.*, sex classification), or other targets associated with protected attribute (Stone et al., 2022; Hashimoto et al., 2018) (*e.g.*, facial attribute classification). It is noteworthy that Type I Bias is predominantly discussed in various biometrics tasks (Conti et al., 2022; Klare et al., 2012; Morales et al., 2020). Compared with various types of targets, protected attributes (*e.g.*, sex, race, and age) are mainly considered the term of attribute in Type I Bias.

---

[1]For instance, Type I Bias involves training sets which yield the long-tail distribution, while Type II Bias typically involves training sets which yields the association between target label and attribute label.

### 4.2 Type II Bias

The general form of Type II Bias is characterized by the dependence between model prediction and attribute. This definition can be used to unify a broad spectrum of papers by considering the status of the attribute and the kind of attribute. The status of the attribute is categorized into three groups: known and labeled, known but unlabeled, and unknown. Specifically, for known and labeled bias, several methods directly leverage attribute labels to explicitly apply a supervision signal for bias mitigation (Zhu et al., 2021). For known but unlabeled bias, several methods mainly utilize the domain knowledge of specific bias attributes to design the module tailored for this bias attribute (Wang et al., 2019a). For unknown bias, several methods identify and emphasize bias-conflicting samples (those exhibiting the opposite bias present in the training set) to mitigate bias (Zhao et al., 2023a). On the other hand, the kind of attribute mainly encompasses sensitive/protected attributes (Angwin et al., 2022b; Chen & Joo, 2021; Calders & Verwer, 2010) and spurious attributes (Nam et al., 2020; Sagawa* et al., 2020; Zhao et al., 2023a). In the case of sensitive attributes, the reliance on them leads to a disproportionate assignment of specific predictions to particular demographic groups, thereby resulting in unfair treatment. In this category, demographic parity (Dwork et al., 2012), a well-known fairness criterion, is often served as a debiasing objective. We present several representative descriptions as follows, *e.g.*,

- *"Demographic parity, which is satisfied when the predictions are independent of the sensitive attributes."* (Creager et al., 2019)

- *"Data fairness can be achieved if the generated decision has no correlation with the generated protected attribute."* (Xu et al., 2018)

- *"Ensuring that the positive outcome is given to the two groups at the same rate."* (Madras et al., 2018)

In the case of spurious attributes, depending on them for decision-making will simplify the training process since models may utilize them as shortcut features instead of learning more comprehensive features during training. However, this leads to model predictions heavily relying on these attributes and further poor generalization performance in real-world applications since such spurious correlation between target and attribute does not generally exist. Several representative descriptions are shown as follows, *e.g.*,

- *"If bias features are highly correlated with the object class in the dataset, models tend to use the bias as a cue for the prediction."* (Hong & Yang, 2021)

- *"Since there are correlations between the target task label and the bias label, the target task is likely to rely on the bias information to fulfill its objective."* (Zhu et al., 2021)

- *"If biased data is provided during training, the machine perceives the biased distribution as meaningful information."* (Kim et al., 2019a)

Table 3: The summary of representative fairness criteria.

| Category | Notion | Definition | Examples |
|---|---|---|---|
| Fairness w.r.t. Type I Bias | Equalized odds (Hardt et al., 2016) | $P(\hat{Y} = y_1\|A = a_0, Y = y) = P(\hat{Y} = y_1\|A = a_1, Y = y), y \in \{y_0, y_1\}$ | Park et al. (2022) Zhang et al. (2023) Conti et al. (2022) |
| | Equal opportunity (Hardt et al., 2016) | $P(\hat{Y} = y_1\|A = a_0, Y = y_1) = P(\hat{Y} = y_1\|A = a_1, Y = y_1)$ | Jung et al. (2022) Yu et al. (2022) Pham et al. (2023) |
| | Accuracy parity (Quan et al., 2023) | $P(\hat{Y} = Y\|A = a_0) = P(\hat{Y} = Y\|A = a_1)$ | Kim et al. (2019b) Zafar et al. (2017a) Quan et al. (2023) |
| Fairness w.r.t. Type II Bias | Demographic parity (Dwork et al., 2012; Kusner et al., 2017) | $P(\hat{Y}\|A = a_0) = P(\hat{Y}\|A = a_1)$ | Creager et al. (2019) Xu et al. (2018) van Breugel et al. (2021) |

Table 4: The overview of the literature regarding Type I Bias and Type II Bias.

| Category | Description | Subsettings | | Examples |
|---|---|---|---|---|
| Type I Bias | Uneven performance of target across attribute | How is performance evaluated? | Error rate | Buolamwini & Gebru (2018) Sattigeri et al. (2019) |
| | | | Loss | Hashimoto et al. (2018) |
| | | | Accuracy | Kim et al. (2019b) |
| | | | Average precision | Ramaswamy et al. (2021) |
| | | | True positive rate | Dhar et al. (2021) Adeli et al. (2021) |
| | | | False positive rate | Xu et al. (2021e) |
| | | | Mean average false rate | Ryu et al. (2017) |
| | | | Confusion matrix | Gong et al. (2020) |
| | | | F1 score | Adeli et al. (2021) |
| | | | Receiver operating characteristic curve (ROC) | Mirjalili et al. (2018) Qin (2020) Yu et al. (2020) |
| | | | Area under the ROC (AUC) | Mirjalili et al. (2019) Gong et al. (2020) Adeli et al. (2021) |
| | | Type of target | Identity | Wang & Deng (2020) Wang et al. (2019b) Gong et al. (2021) |
| | | | Attribute itself | Buolamwini & Gebru (2018) Das et al. (2018) Amini et al. (2019) |
| | | | Other targets associated with protected attribute | Hashimoto et al. (2018) Adeli et al. (2021) Cheng et al. (2021) |
| Type II Bias | Dependence between model prediction and attribute | Is attribute known and labeled? | Known and labeled | Zhu et al. (2021) Ragonesi et al. (2021) Kim et al. (2019a) |
| | | | Known but unlabeled | Wang et al. (2019a) Bahng et al. (2020) Cadene et al. (2019) |
| | | | Unknown | Zhao et al. (2023a) Nam et al. (2020) Jeon et al. (2022) |
| | | Type of attribute | Sensitive attribute/protected attribute | Angwin et al. (2022b) Creager et al. (2019) Madras et al. (2018) |
| | | | Spurious attribute | Sagawa* et al. (2020) Tartaglione et al. (2021) Hong & Yang (2021) |

## 4.3 Fairness Criteria

Besides the papers that explore bias issues directly from the perspective of bias itself, there is another group of papers that leverage established fairness criteria (*e.g.*, demographic parity and equalized odds) as their debiasing objectives. In this section, we first adopt the corresponding definitions of fairness from the definition of bias in Definitions 1 and 2, and then demonstrate that relevant papers based on established fairness criteria can be categorized under these definitions. Given that fairness is the opposite of bias, we can derive the fairness definition for each type of bias as follows,

**Definition 3.** *Fairness w.r.t. Type I Bias. A model f is fair w.r.t.* Type I Bias *if f yields even performance* $d(\hat{Y}, Y)$ *across attribute A,* i.e.*,

$$\sup_{a,a'\in\mathcal{A},d\in\mathcal{M}} |d(\hat{Y}, Y|A = a) - d(\hat{Y}, Y|A = a')| = 0 \tag{3}$$

where $a, a'$ are possible values of $A$ (*e.g.*, female and male), and $\mathcal{M}$ is the set of all potential performance metrics.

**Definition 4.** *Fairness w.r.t. Type II Bias. A model f is fair w.r.t.* Type II Bias *if model prediction $\hat{Y}$ is independent with attribute A,* i.e.*,

$$\sup_{a,a'\in\mathcal{A}} |P(\hat{Y}|A = a) - P(\hat{Y}|A = a')| = 0 \tag{4}$$

where $a, a'$ are possible values of $A$ (*e.g.*, female and male).

Fairness criteria can be categorized into two key classes: group fairness and individual fairness (Mehrabi et al., 2021a; Wang et al., 2022c; Castelnovo et al., 2022). Specifically, group fairness is founded on the idea that "groups of people may face biases and unfair decisions", whereas individual fairness is grounded in the principle that "similar individuals should receive similar decisions" (Castelnovo et al., 2022). We mainly unify group fairness into our definitions since group fairness is more commonly used in fairness research (Du et al., 2020). Group fairness encompasses several well-known fairness criteria such as demographic parity/statistical parity (Dwork et al., 2012; Kusner et al., 2017), equalized odds/equality of odds (Hardt et al., 2016), equal opportunity/equality of opportunity (Hardt et al., 2016), and accuracy parity (Quan et al., 2023). The categorization of them under our fairness definitions is shown in Tab. 3. Specifically, demographic parity, which requires $P(\hat{Y}|A = a_0) = P(\hat{Y}|A = a_1)$, is consistent with Definition 4 when attribute $A$ is binary. Equalized odds, which requires that both even true positive rate (TPR) ($P(\hat{Y} = y_1|Y = y_1)$) and even false positive rate (FPR) ($P(\hat{Y} = y_1|Y = y_0)$) across $A$, and equal opportunity, which is the weaker notion of equalized odds that focuses solely on the advantaged outcome where $Y = y_1$, align with Definition 3 since TPR and FPR are included in the set of performance metrics $\mathcal{M}$. Accuracy parity, where accuracy is represented by $P(\hat{Y} = Y)$, also aligns with Definition 3 since accuracy is the element of $\mathcal{M}$.

## 4.4 Summary

Having unified the prevalent bias issues and well-known fairness criteria under our definitions, in this section, we summarize the main advantages of the proposed definitions. First, the proposed definitions focus on the manifestation of predominant bias, which is clearer and easier to apply compared to definitions based on causes, since the causes of these biases are debatable in some cases (Adeli et al., 2021; Stone et al., 2022; Wang & Deng, 2020). Second, the proposed definitions yield the general form, and by specifying the components in the general form, they can be used to unify a comprehensive list of papers, as summarized in Tab. 4. Third, the proposed definitions, as the first definition to formally define dominant biases, bridge the gap between numerous fairness definitions (Hardt et al., 2016; Kusner et al., 2017; Dwork et al., 2012; Chen et al., 2019; Grgic-Hlaca et al., 2016; Lechner et al., 2021; Quan et al., 2023) and the significant shortage of formal bias definitions. Furthermore, compared with fairness definitions, bias definitions are more practical since encountering bias issues is more common in real-world scenarios, whereas achieving fairness, often considered an ideal benchmark, is rare in practice. Fourth, given that the proposed bias definitions are relatively general, the corresponding fairness definitions are strict, hence aligning with the need for fairness as an ideal standard. Additionally, several well-known fairness criteria can be unified under the proposed fairness definitions.

## 5 Analyzing Confusion between Type I Bias and Type II Bias

In this section, we identify and analyze the widespread confusion in the literature surrounding two predominant types of bias in neural networks: Type I Bias (uneven performance across demographic groups) and Type II Bias (prediction dependence on protected attributes). First, we highlight five major manifestations of confusion. We then explore three underlying causes. Together, these factors explain why the field continues to conflate fundamentally different bias types, hindering progress toward clear and effective fairness solutions.

### 5.1 Manifestations of Confusion

In the previous section, we categorize 415 papers that discuss prevalent biases into two groups based on the manifestation of bias they address. The criteria for this categorization are clearly outlined in Tab. 4. Furthermore, the distinctions between these two types of biases are illustrated in Definitions 1 and 2. However, as summarized in Tab. 5, there is substantial confusion between them in existing literature, which poses challenges for researchers to investigate bias issues. Thus, it is crucial to clarify the confusion and underscore the distinctions between these two types of biases. To this end, in this section, we primarily highlight several prevailing confusions and the potential consequences that arise from overlooking them, based on the investigation of 415 papers.

Table 5: The summary of the existing confusion in the literature regarding bias issues.

| Type of confusion | Examples |
|---|---|
| Ambiguity of Terminology | Wang et al. (2020b); Zhao et al. (2017); Amini et al. (2019) |
| Inaccurate Motivation | Ragonesi et al. (2021); Alvi et al. (2018); Salvador et al. (2022) |
| Lack of Terminology Reuse | Stone et al. (2022); Adeli et al. (2021); Wang & Russakovsky (2023) |
| Abuse of Bias Assessment Metrics | Zhang et al. (2023); Lokhande et al. (2020); Wang et al. (2019c) |
| Weak Existing Distinction | Wang et al. (2022c); Mehrabi et al. (2021a); Le Quy et al. (2022) |

### 5.1.1 Ambiguity of Terminology

One of the confusions is the ambiguity surrounding the terminology of bias. This ambiguity manifests in three primary ways. First, several papers adopt vague terminology such as "bias issues" or simply "bias" without clarifying the particular type of bias they address (Wang et al., 2020b). Furthermore, other commonly used terms such as "model bias" or "algorithmic bias" are also ambiguous, as they might represent either the bias that manifests in the model or the bias that originates from the model itself. Second, studies often denote bias from varied aspects (Hirota et al., 2022; Markl, 2022). For instance, some papers refer to "demographic bias", "gender bias", or "racial bias", emphasizing bias from the perspective of demographic statistics. In contrast, other works utilize "dataset bias", "model bias", or "algorithmic bias", indicating the source of bias. Third, the existing literature frequently uses the same terms to describe different kinds of biases (Liu et al., 2022a; Ragonesi et al., 2021), as summarized in Tab. 6.

**Consequences.** The ambiguity of terminology undermines the clarity of the intended statement and may further lead to misdirected debiasing techniques. For instance, in the abstract of the paper (Zhao et al., 2017), the authors claim that:

- *"We find that (a) datasets for these tasks contain significant gender bias and (b) models trained on these datasets further amplify existing bias."* (Zhao et al., 2017)

In this case, the lack of clarity around the term "gender bias" weakens the significance of the findings. Furthermore, the scope of this ambiguity is extensive. Specifically, sections including "Title", "Abstract", "Introduction", and "Related Work" are often impacted, as there may lack sufficient context for a precise interpretation (Sadeghi et al., 2019; Gordaliza et al., 2019). More concerned, the vagueness may persist throughout the entire paper (Amini et al., 2019) if the addressed bias is not disambiguously clarified in the "Problem Statement" or evaluation protocol in the "Experiments" section.

Table 6: The summary of terms commonly used for bias.

| Paper | Claimed bias to address (Motivation) | Actual type of bias to address (Technique) | |
|---|---|---|---|
| | | Type I Bias | Type II Bias |
| Wang et al. (2019b) | Racial bias | ✓ | |
| Dhar et al. (2021) | Gender bias, skintone bias | ✓ | ✓ |
| Conti et al. (2022) | Gender bias | ✓ | ✓ |
| Wang et al. (2019c) | Gender bias | | ✓ |
| Zhao et al. (2017) | Gender bias | | ✓ |
| Wang et al. (2020b) | Gender bias | | ✓ |
| Amini et al. (2019) | Algorithmic bias | ✓ | |
| Liu et al. (2022a) | Dataset bias | ✓ | |
| Adeli et al. (2021) | Dataset bias | ✓ | |
| Ragonesi et al. (2021) | Dataset bias | | ✓ |
| Lee et al. (2021) | Dataset bias | | ✓ |

### 5.1.2 Inaccurate Motivation

Another confusion is that existing work addressing these two types of bias inaccurately cites each other for their own motivation. For instance, some studies (Ragonesi et al., 2021; Alvi et al., 2018) that address Type II Bias motivate themselves by the uneven performance in face recognition, a manifestation of Type I Bias. Other work (Wang & Deng, 2021; Salvador et al., 2022) that tackles Type I Bias in debiasing face recognition is motivated by the correlation between model predictions and spurious attributes in facial attribute classification (Alvi et al., 2018), a manifestation of Type II Bias. Furthermore, this confusion is aggravated as some papers are motivated by semi-relevant work. Specifically, as highlighted by (Grother et al., 2019), debiasing face recognition literature (Salvador et al., 2022; Wang et al., 2019b; Wang & Deng, 2020) tends to be motivated by the manifestation of worse accuracy for minority groups in sex classification (Buolamwini & Gebru, 2018), rather than the direct issue of uneven performance in face recognition (Robinson et al., 2020; Pahl et al., 2022).

**Consequences.** Inaccurate motivation leads to misunderstanding and misalignment in the existing literature. Furthermore, this issue may compound over time, as the subsequent work built upon the papers with such inaccurate motivation will perpetuate the confusion.

### 5.1.3 Lack of Terminology Reuse

The confusion also manifests in the introduction of new terms in different papers addressing the same bias. For instance, "minority group bias" (Stone et al., 2022), "dataset bias" (Adeli et al., 2021), and "bias as underrepresentation" (Wang & Russakovsky, 2023) are all used to denote uneven performance across attributes (Type I Bias).

- *"Dataset bias is often introduced due to the lack of enough data points spanning the whole spectrum of variations with respect to one or a set of protected variables."* (Adeli et al., 2021)

- *"Minority group bias. When a subgroup of the data has a particular attribute or combination of attributes that are relatively uncommon compared to the rest of the dataset, they form a minority group. A model is less likely to correctly predict for samples from a minority group than for those of the majority."* (Stone et al., 2022)

- *"[...] 'bias' means that one appearance of an object is underrepresented."* (Wang & Russakovsky, 2023)

Similarly, "sensitive attribute bias" (Stone et al., 2022), "task bias" (Adeli et al., 2021), and "bias as spurious correlation" (Wang & Russakovsky, 2023) all signify the dependence between model prediction and attribute (Type II Bias).

- *"Task bias, on the other hand, is introduced by the intrinsic dependency between protected variables and the task."* (Adeli et al., 2021)

- *"Sensitive attribute bias. A sensitive attribute (also referred to as "protected") is one that should not be used by the model to perform the target task, but which provides an unwanted "shortcut" that is easily learned, and results in an unfair model."* (Stone et al., 2022)

- *"[...] considering bias in the form of spurious correlations between the target label and a sensitive attribute which is predictive on the training set but not necessarily so on the test set."* (Wang & Russakovsky, 2023)

**Consequences.** These inconsistent definitions can further contribute to confusion, with some highlighting the manifestation of the bias while others delving into the underlying causes of the bias. Furthermore, without a unified terminology for the predominant biases, it becomes challenging to systematically gather and compare relevant work.

### 5.1.4   Abuse of Bias Assessment Metrics

The usage of bias assessment metrics exhibits the confusion in two primary ways. First, the bias assessment metrics, which are designed independently of debiasing methods, are rarely used (Li & Abd-Almageed, 2021; Wang & Russakovsky, 2021). Instead, many works tend to introduce their own metrics to demonstrate the effectiveness of the proposed debiasing method (Wang et al., 2019c; Zhao et al., 2017), which leads to an overwhelming number of metrics. Second, some studies inappropriately employ indirect bias assessment metrics or even metrics that are not designed for the specific bias they address. For instance, several studies (Zhang et al., 2023; Lokhande et al., 2020) motivated by the dependence between model prediction and attributes (the manifestation of Type II Bias) use true positive rate (TPR) difference and false positive rate (FPR) difference for evaluation. However, as highlighted by (Wang & Russakovsky, 2021), metrics such as TPR difference, FPR difference, accuracy difference, and average mean-per-class accuracy difference, are not suitable for evaluating Type II Bias since they fail to consider the dependence between target and attribute in the training set and cannot distinguish between an increase or decrease of dependence in learned representation.

**Consequences.** The abuse of bias assessment metrics leads to inaccurate evaluations of debiasing performance in relation to the specific type of bias being addressed, hence exacerbating confusion in the field. Furthermore, it also complicates the comparison between different debiasing methods and hinders the construction of a unified evaluation protocol.

### 5.1.5   Weak Existing Distinction

Despite the evident confusion in the literature, numerous studies, especially survey papers, have not sufficiently distinguished Type I Bias and Type II Bias. Furthermore, the confusion is not only widespread but has also persisted for a significant duration, as shown by the timeframes of the investigated papers. However, the bias taxonomy, presented in surveys over time (Wang et al., 2022c; Mehrabi et al., 2021a; Le Quy et al., 2022), may fail to clearly differentiate between these two types of biases. Alarmingly, a recent and highly cited survey on machine learning bias (Mehrabi et al., 2021a) scarcely cites papers that discuss Type II Bias stemming from spurious correlations between target and attribute, thereby overlooking the distinction from Type I Bias.

**Consequences.** The weak distinction between these two types of biases in existing surveys will exacerbate the prevailing confusion in this field over time. Consequently, due to the lack of clarity, which surveys were originally designed to provide concerning the categorization of bias issues, these bias issues will eventually be undesirably conflated.

## 5.2   Underlying Causes of Confusion

In this section, we investigate various factors that may contribute to the confusion discussed in the previous section. Specifically, we examine the historical context, the preconception about bias, and the methodologies adopted to address different biases, to provide insights on how and why such confusion has persisted in the literature.

### 5.2.1   Historical Context

We first examine the historical origins of bias issues. In Fig. 2, we summarize the enrichment of the concept "bias" in machine learning from the perspective of Type I Bias and Type II Bias and highlight key milestones throughout its history. Originally, "bias" is defined as unfair favoritism or prejudice towards one thing, person, or group over another (DiTomaso, 2015). Specifically, bias issues are especially evident in real-world decision-making processes, such as advertising, financial creditworthiness, employment, education, and criminal justice (Ruggeri et al., 2023; Edmond & Martire, 2019). To promote fairness, certain sensitive attributes (*e.g.*, sex, age, and race) are by law defined as protected attributes that cannot be discriminated against in the decision-making process (Corbett-Davies & Goel, 2018). In this initial stage, decisions are primarily made by humans. Thus, the main bias issue is whether human decision-making depends on protected attributes, which aligns with Type II Bias in our definitions.

| Trustworthy AI | | | |
|---|---|---|---|
| Subject | Human | Machine intelligence | Machine intelligence |
| Bias manifestation | Does prediction depend on attribute? | Does prediction depend on attribute? | Is performance uneven across attribute? |
| Category | Type II Bias | Type II Bias | Type I Bias |

2012      2018

Dwork et al., "Fairness through awareness", 2012    Buolamwini & Gebru, "Gender shades: Intersectional accuracy disparities in commercial gender classification, 2018

Figure 2: The enrichment of the concept "bias" in machine intelligence with important milestones. Initially, "bias" implied that human decision-making depends on protected attributes (Type II Bias). As machine intelligence began aiding human decision-making processes, the subject of "bias" broadened from humans to algorithms. Along with the continued advances of machine intelligence, a new aspect of bias issues, performance disparity across demographic groups (Type I Bias), further enriched the meaning of "bias". Currently, addressing both Type I Bias and Type II Bias is essential for ensuring Trustworthy AI.

Following the emergence of neural networks, machine learning models start to assist in human decision-making processes (Bastani et al., 2021; Dankwa-Mullan et al., 2019). This evolution also leads to an expansion of the subject in the discussion regarding bias issues, from human decision-making to algorithmic decision-making (Starke et al., 2022). With this change, numerous works begin to explore if algorithmic decision-making depends on protected attributes (*i.e.*, demographic parity) (Dwork et al., 2012; Kusner et al., 2017), which also aligns with Type II Bias. Meanwhile, along with the advancement of neural networks, their performance becomes a crucial evaluation criterion. Consequently, it brings significant attention to a new aspect of bias issues: performance disparity across demographic groups (Buolamwini & Gebru, 2018; Quan et al., 2023), which aligns with Type I Bias in our definitions. Furthermore, new fairness criteria such as equalized odds and equal opportunity (Hardt et al., 2016), which address disparities in true positive rates and false positive rates across demographic groups, are adopted from demographic parity.

We conjecture that the confusion arises because the term "bias" in neural networks has been endowed with multiple important meanings over time without well-defined distinctions. This ambiguity leads individuals to interpret different types of predominant biases from the same term. Specifically, some individuals associate the primary bias with performance disparity due to the critical role of model performance in model evaluation. Conversely, other individuals prioritize prediction disparity since it is the prevalent bias deeply embedded in real-world scenarios. Consequently, denoting these two different but predominant biases with the single term "bias" results in misunderstandings in the broader literature.

### 5.2.2 Preconception about Bias

The preconception of researchers about bias, stemming from their specific relevant fields, also contributes to the confusion. Specifically, bias issues encompass a wide range of relevant fields, some of which are associated with Type I Bias and others with Type II Bias. For instance, Type I Bias involves long-tail distribution (Cao et al., 2020), catastrophic forgetting (Kirkpatrick et al., 2017), domain adaptation (Li et al., 2014), and various biometric tasks (Xiao et al., 2023; Hutiri & Ding, 2022). In contrast, Type II Bias involves shortcut learning (Geirhos et al., 2020), simplicity bias (Teney et al., 2022), invariant representation learning (Creager et al., 2019), out-of-distribution challenges (Shen et al., 2021). In this sense, researchers from diverse fields hold their own preconceived notions of bias based on their field-specific knowledge. For instance, in several biometric tasks (*e.g.*, face recognition, face detection, face verification) With identity as target and sex as an attribute, uneven performance across sex (the manifestation of Type I Bias) is naturally regarded as bias since the primary focus of biometric systems is on model performance (Robinson et al., 2020).

However, the dependence between model prediction and attribute (the manifestation of Type II Bias) might not be considered as bias since there naturally only exists non-overlapping targets across attribute (Wang et al., 2019b). For instance, an individual can be categorized as either male or female but not both, thereby resulting in a natural association between identity prediction and specific sex. Furthermore, due to the absence of clear distinctions regarding bias issues, research groups from different fields may not share a unified perspective on bias and may interpret it differently. However, they use similar bias-related terms in their papers and present them in the same venues, which potentially causes confusion regarding bias issues.

### 5.2.3 Similar Methodologies

The existing confusion also arises from the overlap in methodologies used to address Type I Bias and Type II Bias. For instance, to mitigate Type I Bias, several studies (Morales et al., 2020; Gong et al., 2020; Dhar et al., 2021) enhance the performance for minority groups by preventing the model from encoding the information of protected attributes. Similarly, to tackle Type II Bias, some methods (Ragonesi et al., 2021; Zhu et al., 2021; Kim et al., 2019a) aim to develop representations that are invariant to the protected attribute by minimizing mutual information between the learned representation and the protected attribute. Both of these methods can be categorized into invariant representation learning (Arjovsky et al., 2019). Furthermore, domain adaptation is also utilized for both Type I Bias (Kan et al., 2015; Guo et al., 2020) and Type II Bias (Rosenfeld et al., 2022). These similarities in methodologies obscure the distinction between Type I Bias and Type II Bias, thereby inducing confusion.

## 6 Experimental Discussion

In this section, we empirically investigate the distinction between Type I Bias and Type II Bias. Specifically, we conduct experiments on two synthetic datasets and two well-known real-world datasets: Adult Income Dataset (Dua & Graff, 2017) and CelebA Dataset (Liu et al., 2015). First, we use synthetic data to demonstrate that Type I Bias and Type II Bias are unrelated, *i.e.*, one can exist without the presence of the other bias. Next, we utilize the Adult dataset to further illustrate the difference between Type I Bias and Type II Bias in real-world scenarios. Last, we employ the CelebA dataset to evaluate the effectiveness of multiple representative bias assessment metrics in assessing Type I Bias and Type II Bias. All experimental results are obtained by averaging the results over 10 trials. While our experimental validation focuses on image and tabular data, the proposed taxonomy and definitions extend naturally to language tasks. For example, Type I Bias in text classification can manifest as uneven performance across demographic subgroups, such as sentiment classifiers that perform better on reviews written in Standard American English than on African American Vernacular English, even when ground truth labels are consistent (Tatman, 2017). Besides, Type II Bias in text tasks often arises when model predictions (*e.g.*, toxicity, hate speech, or occupation classification) are influenced by identity-indicative features such as names, pronouns, or phrases associated with a protected group—even when these features are irrelevant to the task (Gallegos et al., 2024).

### 6.1 Unrelated Occurrence

In this section, we leverage synthetic data to simulate two scenarios: the first scenario showcases the presence of Type I Bias without Type II Bias, while the second scenario showcases the presence of Type II Bias without Type I Bias.

**Setup.** We construct the synthetic dataset containing instances $(x, y)$, where $x$ denotes a two-dimensional input consisting of the useful feature $u$ and the binary attribute $a$, and $y$ denotes the target label. Next, we apply a classifier $C : \mathcal{X} \to \mathcal{Y}$ to consume the input $x$ and produce the prediction $\hat{y} = C(x) = C(u, a) \in \mathcal{Y}$. The classifier is a single fully connected layer (FC) followed by the binary cross-entropy loss. To evaluate Type I Bias, we measure the difference in accuracy. To assess Type II Bias, we utilize the Calders-Verwer discrimination score (Calders & Verwer, 2010) defined as $|P(\hat{Y} = y|A = 1) - P(\hat{Y} = y|A = -1)|$.

### 6.1.1 Type I Bias Exists without Type II Bias

To induce Type I Bias, we construct a training set which is imbalanced across attribute $A$, where the subset with $A = -1$ represents the minority group, as shown in Fig. 3. Additionally, we vary the optimal classification boundary across $A$ since one widely accepted cause of Type I Bias is that the model trained on the sufficient samples in majority groups might not effectively generalize to minority groups (Wang & Russakovsky, 2023). And, we construct a testing set which is balanced across values of the attribute $A$. Details of the dataset construction are presented in Appendix 2.2.

**Analysis.** In Fig. 3, we observe that the learned classification boundary is vertical at $X = 0$, which is primarily determined by dominant samples in the majority group. The vertical boundary suggests that the model does not use attribute $A$ for classification. Furthermore, as highlighted in Tab. 7, given that $P(\hat{Y} = y | A = 1) = P(\hat{Y} = y | A = -1) \ \forall \ y \in \{0, 1\}$, model prediction $\hat{Y}$ is independent with attribute $A$, *i.e.*, Type II Bias does not exist. However, it is noteworthy that there is a significant performance disparity between the majority and minority groups, which confirms the existence of Type I Bias.

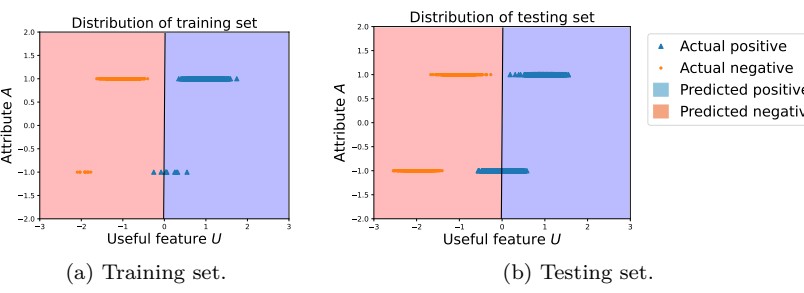

(a) Training set.     (b) Testing set.

Figure 3: Distribution of training and testing sets regarding synthetic data. The vertical classification boundary (labeled as the black line) reveals that the classifier does not utilize $A$ for classification. However, there are more wrong predictions in the group of $A = -1$ than in the group of $A = 1$, which violates performance parity.

Table 7: Type I Bias exists without Type II Bias since there exists accuracy disparity across $A$ while $\hat{Y}$ and $A$ are independent.

|  | Accuracy | $P(\hat{Y} = 0 | A)$ | $P(\hat{Y} = 1 | A)$ |
|---|---|---|---|
| $A = 1$ | 100.00 | 66.7% | 33.3% |
| $A = -1$ | 65.33 | 66.7% | 33.3% |
| $|\Delta|$ | 34.67 | 0 | 0 |

### 6.1.2  Type II Bias Exists without Type I Bias

To induce Type II Bias, we construct a training set in which the combinations $(A = 1, Y = 0)$ and $(A = -1, Y = 1)$ occur more frequently than other combinations, as illustrated in Fig. 4. This setup is motivated by the widely accepted understanding that a spurious association between the target label $Y$ and the attribute $A$ in the training data is a key contributor to Type II Bias (Nam et al., 2020; Zhu et al., 2021; Tartaglione et al., 2021). For evaluation, we construct a testing set that is balanced across both $Y$ and $A$, ensuring no association between the two variables. Details of the dataset construction are presented in Appendix 2.2.

**Analysis.** In Fig. 4, we observe that the learned classification boundary is not vertical, which suggests that the classifier relies on $A$ for decision-making. Furthermore, as highlighted in Tab. 8, given that $P(\hat{Y} = y | A = 1) \neq P(\hat{Y} = y | A = -1) \ \forall \ y \in \{0, 1\}$, model prediction $\hat{Y}$ is not independent with attribute $A$, *i.e.*, Type II Bias exists. However, for Type I Bias, it is noteworthy that there is no significant performance disparity between the majority and minority groups.

### 6.2  Different Manifestations in the Real World

In this section, we utilize the Adult Income Dataset (Dua & Graff, 2017) to illustrate different manifestations of Type I Bias and Type II Bias in real-world scenarios. Adult Dataset is a census dataset where the target is whether a person earns a higher income (over 50K USD per year), and the protected attribute is sex. As shown in Tab. 9, the dataset is partitioned into four quarters based on the combination of target labels and protected attribute labels, given that both are binary in nature. The statistics illustrate that the Adult dataset is well-suited for investigating both Type I Bias and Type II Bias. Specifically, the dataset exhibits an uneven distribution across sex, with a larger number of female individuals (16,192) compared to male individuals (32,650), which could induce Type I Bias. Furthermore, the dataset also exhibits a substantial

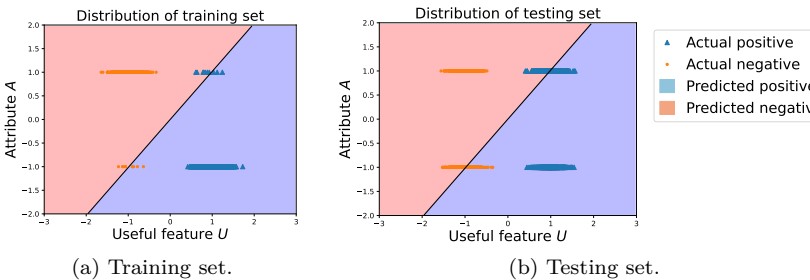

(a) Training set.  (b) Testing set.

Table 8: Type II Bias exists since $\hat{Y}$ and $A$ are not independent while there is no accuracy disparity across $A$.

|  | Accuracy | $P(\hat{Y}=0|A)$ | $P(\hat{Y}=1|A)$ |
|---|---|---|---|
| $A=1$ | 85.98 | 64.1% | 35.9% |
| $A=-1$ | 85.97 | 35.4% | 64.6% |
| $|\Delta|$ | $\approx 0$ | 28.7% | 28.7% |

Figure 4: Distribution of training and testing sets regarding synthetic data. The non-vertical classification boundary (labeled as the black line) reveals that the classifier utilizes $A$ for classification. However, the number of wrong predictions is approximately the same across $A$, thereby fulfilling performance parity.

disparity in the number of samples with higher income between females (1,769) and males (9,918), which could induce Type II Bias. Besides, we consider the magnitude of disparity as a reflection of bias strength. In practice, we acknowledge that the significance of a given disparity is inherently context-dependent. So, it is advisable in practical deployments to apply domain-specific thresholds when interpreting bias metrics.

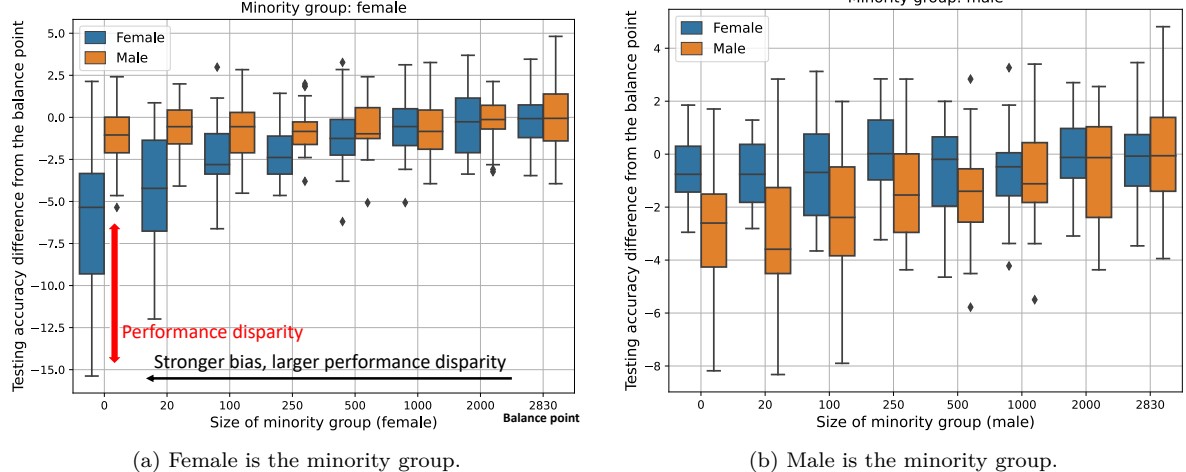

(a) Female is the minority group.  (b) Male is the minority group.

Figure 5: Illustration of Type I Bias on Adult, which manifests as uneven performance between the minority and majority groups. As Type I Bias becomes stronger (the minority size decreases), the accuracy for the minority group diminishes while the accuracy for the majority group remains unchanged, thereby enlarging the performance disparity across the minority and majority groups.

**Setup.** We perform data pre-processing on the input census data. Specifically, we transform the categorical features using one-hot encoding and normalize the numerical features into a Gaussian distribution with zero mean and unit variance. Consequently, each input sample is transformed into a 108-dimensional vector. For the training model, we employ a three-layer multilayer perceptron (MLP) followed by the binary cross-entropy loss as the baseline classifier.

### 6.2.1 Type I Bias

To investigate Type I Bias, we construct several imbalanced training sets and control the bias strength by modifying the degree of imbalance in the training set. Specifically, we initially construct a balanced training

Table 9: Statistics of the Adult dataset. The number of females is greater than the number of males, which could induce Type I Bias. Furthermore, the number of samples with higher income and samples with lower income are different across sex categories, which could induce Type II Bias.

|  | Higher income | Lower income | Total |
|---|---|---|---|
| Female | 1,769 | 14,423 | 16,192 |
| Male | 9,918 | 22,732 | 32,650 |
| Total | 11,687 | 37,155 | 48,842 |

set across both target $Y$ and attribute $A$ using 80% of the entire dataset and a balanced testing set with the remaining samples. We then manually adjust the size of the minority group in the training set while maintaining the size of the majority group to control bias strength. Additionally, we construct two distinct groups of training sets, with either females or males as the minority group. For instance, considering the setting where the female is a minority group and the minority size is 100, the training set would consist of 50 higher-income females and 50 lower-income females, in addition to all males from the balanced training set. We conduct experiments under different minority sizes and present the testing performance versus the size of the minority group in Fig. 5.

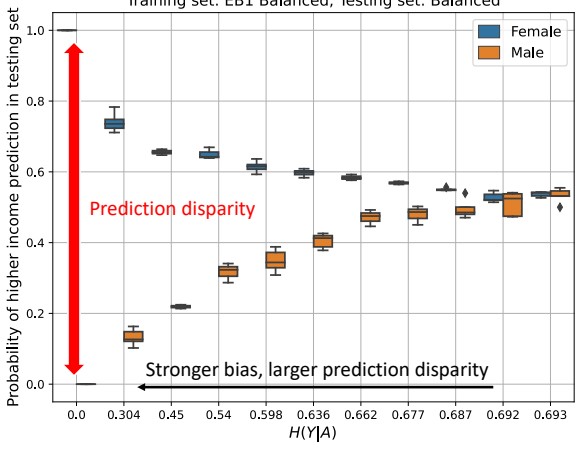
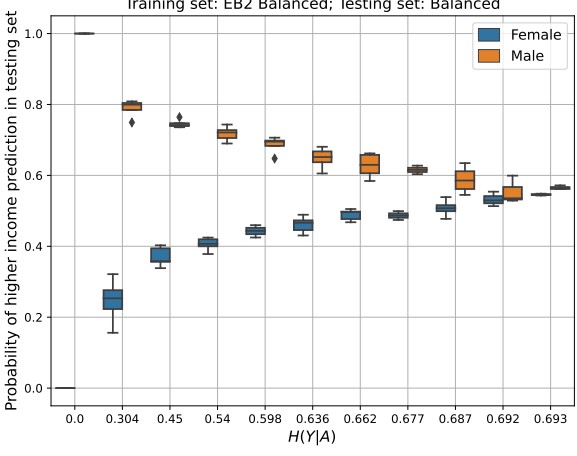

(a) Trained on EB1 Balanced consisting of females with higher income and males with lower income.

(b) Trained on EB2 Balanced consisting of females with lower income and males with higher income.

Figure 6: Illustration of Type II Bias on Adult, which manifests as the dependence between model prediction and attribute. As Type II Bias intensifies ($H(Y|A)$ decreases, rendering the attribute more predictable of the target), the prediction probability in outputting a specific prediction diverges between females and males, *i.e.*, decision-making increasingly relies on the attribute.

**Analysis.** Notably, we notice a non-zero accuracy disparity between females ($85.15\%_{\pm 1.52}$) and males ($78.38\%_{\pm 1.90}$) at the balance point where the training set is evenly distributed across both target $Y$ and attribute $A$. We conjecture that this disparity is mainly because certain groups are inherently more difficult to classify than other groups (Klare et al., 2012). To facilitate a clearer analysis of Type I Bias, we use the accuracy difference from the testing accuracy at the balance point to represent the testing performance. This difference in testing accuracy, denoted as $Acc_{\text{diff}}$, is calculated by subtracting the testing accuracy at the balance point from the absolute accuracy at a given bias strength, *i.e.*, $Acc_{\text{diff}} = Acc_{\text{abs}} - Acc_{\text{balance}}$. In Fig. 5, we observe that the performance disparity exists across the minority group and the majority group. The accuracy for the minority group tends to decrease as its size diminishes (bias strength increases), especially when there are very limited samples from the minority group. Furthermore, in Fig. 5a, we observe that stronger bias results in larger performance fluctuations (bigger spread in the boxplot), which highlights the lack of robustness under such conditions. In summary, the manifestation of Type I Bias in

real-world scenarios is uneven performance across demographic groups. One plausible cause is the imbalance in data representation across these groups in the training set. For instance, some demographic groups may be underrepresented due to long-tail distribution (Cao et al., 2020), resulting in a skewed distribution of samples across different demographic groups. Consequently, while data-driven models are more accurately trained on demographic groups with sufficient samples, they may not be as effective for underrepresented groups, which leads to poor prediction accuracy and unfairness towards these groups.

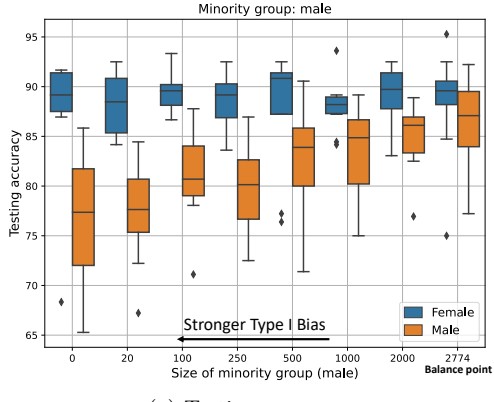

(a) Testing accuracy.

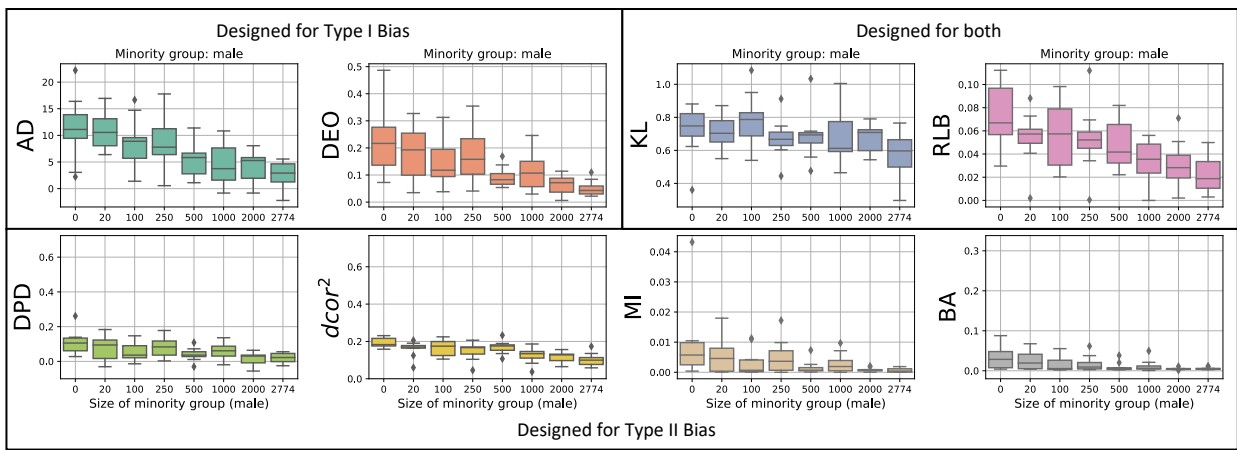

(b) Evaluation with various bias assessment metrics.

Figure 7: Investigation of Type I Bias on CelebA with males as the minority group. As bias strength diminishes (the size of the minority group enlarges), the accuracy of the minority group enhances, leading to a reduction in the accuracy disparity between females and males, and the bias assessed by metrics tailored to evaluate Type I Bias is also mitigated.

### 6.2.2 Type II Bias

To investigate Type II Bias, we construct the training set where the target $Y$ is associated with the attribute $A$ and control the bias strength by adjusting the strength of the association between $Y$ and $A$ in the training set. Specifically, we initially construct two balanced training datasets consisting of 3538 records, each associating either females or males with higher income: (1) Extreme Bias 1 Balanced (EB1 Balanced) only contains females with higher income and males with lower income, and (2) Extreme Bias 2 Balanced (EB2 Balanced) only contains males with higher income and females with lower income. Subsequently, we adjust the percentage of bias-conflicting samples (samples with the opposite bias present in the training set) while ensuring a consistent number of biased samples. This strategy enables us to construct multiple training sets, each with a distinct conditional entropy $H(Y|A)$ (*i.e.*, the smaller $H(Y|A)$, the more predictive the attribute $A$ is of the target $Y$, and the stronger the bias). Additionally, we construct a balanced testing

set (Balanced) consisting of 7076 records, ensuring an even distribution of all combinations of target and attribute labels. Note that all these datasets are designed to be balanced across attributes to mitigate the effect of Type I Bias.

**Analysis.** In Fig. 6, we observe that there is a significant prediction disparity between females and males. Furthermore, this disparity becomes more pronounced as $H(Y|A)$ diminishes (the bias strength increases). In summary, the manifestation of Type II Bias in real-world scenarios is the dependence on the attribute in decision-making processes. One widely accepted reason is an uneven distribution of *specific target groups* across attributes, distinguishing it from Type I Bias, which emerges from an uneven distribution of samples across attributes. For instance, the collected dataset may contain more negative samples for female individuals and positive samples for male individuals compared to other target-attribute combinations. During training, the model may leverage sex as a shortcut feature to simplify the learning process, rather than learning more comprehensive features. However, such an association between specific targets and attributes does not generally exist in the real world. Consequently, during applying, the trained model may still rely on the attribute, which leads to a higher frequency of positive outcomes for specific individuals and further unfair treatment for these groups.

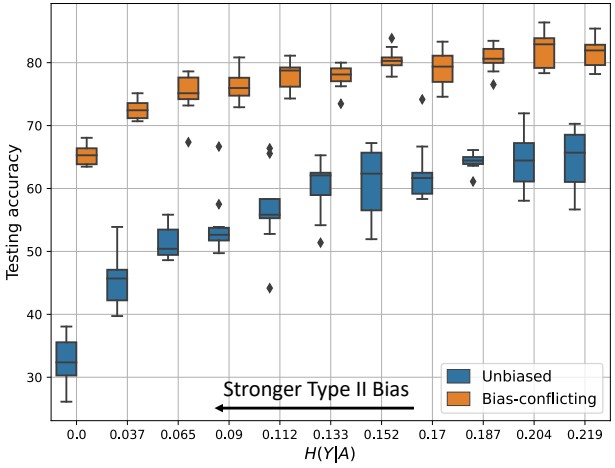

(a) Testing accuracy.

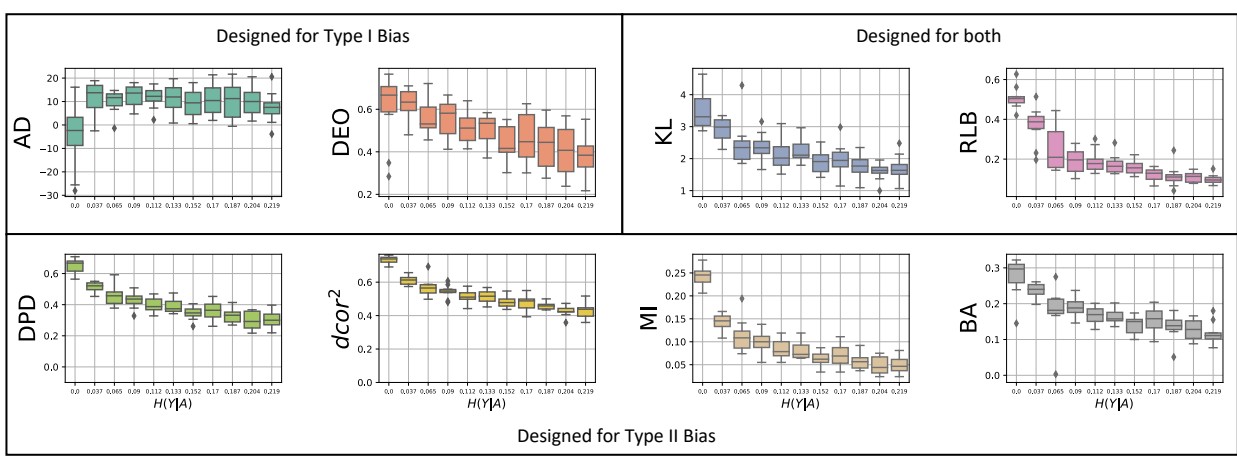

(b) Evaluation with various bias assessment metrics.

Figure 8: Investigation of Type II Bias on CelebA. The evaluation of bias assessment metrics is conducted on *unbiased* testing set. As bias strength diminishes ($H(Y|A)$ increases, rendering the attribute less predictive of the target), the accuracies of both *unbiased* and *bias-conflicting* enhance, and the bias assessed by metrics tailored to evaluate Type II Bias is also mitigated.

### 6.2.3 Summary

As shown in Fig. 5, Type I Bias manifests as the performance disparity across $A$, which is evaluated based on the joint distribution of model prediction $\hat{Y}$ and ground truth $Y$. Conversely, as shown in Fig. 6, Type II Bias manifests as the prediction disparity across $A$, which is evaluated solely based on the distribution of model prediction $\hat{Y}$. Thus, Type I Bias and Type II Bias are unrelated phenomena and exhibit different impacts on the fairness of neural networks.

## 6.3 Evaluation of Various Metrics

In this section, we employ the CelebA dataset (Liu et al., 2015) to investigate several representative bias assessment metrics in assessing Type I Bias and Type II Bias. CelebA dataset is an image dataset of human faces where facial attributes (*e.g.*, blond hair) are the prediction target $Y$ and sex is the attribute $A$. As illustrated in Tab. 10, the CelebA dataset is divided into four parts based on all possible combinations of the binary target and protected attribute labels. The statistics suggest the presence of both Type I and Type II biases in the CelebA dataset.

Table 10: Statistics of the CelebA dataset. The number of females is greater than the number of males, which could induce Type I Bias. Furthermore, the number of samples with blond hair and samples without blond hair are different across sex categories, which could induce Type II Bias.

|  | Blond hair | Non-blond hair | Total |
|---|---|---|---|
| Female | 28,234 | 89,931 | 118,165 |
| Male | 1,749 | 82,685 | 84,434 |
| Total | 29,983 | 172,616 | 202,599 |

**Setup.** To construct training and testing sets, we follow the setup of Adult explained above. In the case of Type I Bias, we construct several training sets with varying bias strength by modifying the size of the minority group in the training set. For testing, we construct a testing set that is balanced across both target and attribute. In the case of Type II Bias, we construct training sets where facial attributes are associated with a particular sex. Specifically, we construct an extreme bias version of the training set consisting of 89754 images with $H(Y|A) = 0$, denoted *TrainEx*, where the bias-conflicting samples (samples exhibiting the opposite bias in the training set) are removed from the original training set. Furthermore, we control bias strength by adjusting the proportion of bias-conflicting samples while maintaining the number of biased samples (samples exhibiting the same bias observed in the training set). For testing, we construct two testing sets: (1) *Unbiased*, consisting of 720 images which contain an even number of samples across all combinations of target and attribute, and (2) *Bias-conflicting*, consisting of 360 images where all biased samples are excluded from *Unbiased* testing set (only bias-conflicting samples remain). In both studies, we consider *blond hair* as the prediction target. For the training model, we utilize ResNet18 (He et al., 2016) followed by the binary cross-entropy loss as the baseline classifier without any debiasing techniques. For bias assessment, we employ a comprehensive list of representative metrics including accuracy disparity (AP) (Quan et al., 2023), difference in equality of opportunity (DEO) (Morales et al., 2020), KL-divergence between score distributions (KL) (Chen & Wu, 2020), representation-level bias (RLB) (Li & Abd-Almageed, 2021), demographic parity distance (DPD) (Creager et al., 2019), distance correlation ($dcor^2$) (Székely et al., 2007), mutual information (MI) (Li & Abd-Almageed, 2023), and bias amplification (BA) (Zhao et al., 2017; Wang & Russakovsky, 2021).

**Analysis.** In the case of Type I Bias, as shown in Fig. 7a, there exists a noticeable performance disparity across sex. As the size of the minority group increases (bias strength diminishes), the performance of the minority group improves, and the performance gap between the minority and majority groups is mitigated. Notably, the performance gap is nonzero even at the balance point, with females achieving higher accuracy than males. We hypothesize that this is because blond hair is more visually prominent in females with long hair. Consequently, even if the dataset is balanced across sex, males may still be relatively underrepresented, *i.e.*, male images are still insufficient for the model to learn a robust representation of males. In the case of

Type II Bias, as shown in Fig. 8a, the testing accuracy of both *Unbiased* and *Bias-conflicting* testing sets rises as $H(Y|A)$ increases (bias strength diminishes).

For the evaluation of various bias assessment metrics, in Figs. 7b and 8b, we observe a noticeable decline in the metrics tailored for a specific type of bias as the corresponding bias strength diminishes. It is noteworthy that the mean of accuracy disparity (AD) approaches zero in the extreme bias case of Type II Bias, where $H(Y|A) = 0$ (the leftmost point). This can be attributed to the fact that, in such extreme bias situations, the target label is bijectively mapped to the attribute label in the training set. Consequently, the trained model may output arbitrary predictions for both sex in the testing set, which leads to an accuracy disparity that is nearly zero.

# 7 Path to Follow

In this section, we present a more comprehensive comparison between Type I Bias and Type II Bias based on our investigation of 415 papers. Our comparison encompasses multiple aspects, including the underlying causes, debiasing methods, evaluation protocol, prevalent datasets, and future directions. Most notably, for each type of bias, we summarize debiasing methods in Tab. 11, bias assessment metrics in Tab. 12, and prevalent datasets in Tabs. 13 and 14.

We use classification as a case study due to the abundance of existing work in this area. Nevertheless, the proposed definitions are generalizable to tasks beyond classification. For instance, in regression tasks, mean squared error can be used to assess Type I Bias, while standard measures of attribute dependence (*e.g.*, mutual information) can be applied to evaluate Type II Bias. We hope the comparison can alleviate the cognitive burden from the prevailing confusion between these two types of biases and serve as a roadmap for new researchers to follow.

Table 11: The summary of debiasing methods.

| Category | Pre-processing | In-processing | Post-processing |
|---|---|---|---|
| Type I Bias | Balanced dataset collection Buolamwini & Gebru (2018); Karkkainen & Joo (2021); Synthetic dataset generation Balakrishnan et al. (2021); Li & Abd-Almageed (2023); Strategic sampling or reweighting Wang & Deng (2020) | Domain adaptation Wang et al. (2019b); Guo et al. (2020); Kan et al. (2015); Attribute removal Gong et al. (2020); Dhar et al. (2021) | Calibrated equalized odds Pleiss et al. (2017) |
| Type II Bias | Universal dataset collection Li et al. (2023a); Synthetic dataset generation Ramaswamy et al. (2021); Sattigeri et al. (2019); Domain randomization Tobin et al. (2017) | Mutual information minimization Kim et al. (2019a); Ragonesi et al. (2021); Zhu et al. (2021); Domain-invariant learning Sagawa* et al. (2020); Ahmed et al. (2021); Creager et al. (2021); Adversarial training Nam et al. (2020); Alvi et al. (2018); Zhang et al. (2018) | Ensemble domain-independent training Wang et al. (2020b) |

Table 12: The summary of bias assessment metrics.

| Category | Metrics |
|---|---|
| Type I Bias | Difference in performance evaluated by various criteria (*e.g.*, accuracy disparity (AD) Kim et al. (2019b); Quan et al. (2023); Zafar et al. (2017a); Zhao et al. (2019a)) |
| | Difference in equality of opportunity (DEO) Morales et al. (2020); Quadrianto et al. (2019); Sattigeri et al. (2019); Lokhande et al. (2020); Ramaswamy et al. (2021) |
| | Equal error rate (EER) Mirjalili et al. (2019) |
| Type II Bias | Demographic parity distance (DPD) Creager et al. (2019); Kim et al. (2019b); Sattigeri et al. (2019) |
| | Distance correlation ($dcor^2$) Székely et al. (2007); Adeli et al. (2021) |
| | Mutual information (MI) Li & Abd-Almageed (2023) |
| | Bias amplification (BA) Wang et al. (2020b); Ramaswamy et al. (2021), Directional BA Wang & Russakovsky (2021); Ramaswamy et al. (2021), Multi-attribute BA Zhao et al. (2023c) |
| | Disparity impact Zafar et al. (2017b); Bellamy et al. (2019) |
| | Representation bias Li et al. (2018); Li & Vasconcelos (2019) |
| | Logit-level loss Xie et al. (2017); Jaiswal et al. (2018) |
| Both | KL-divergence between score distributions (KL) Chen & Wu (2020); Ramaswamy et al. (2021) |
| | Representation-level bias (RLB) Li & Abd-Almageed (2021) |

## 7.1 Type I Bias

### 7.1.1 Underlying Causes

Data imbalance across different demographic groups in the training set is commonly accepted as the possible cause for Type I Bias (Cherepanova et al., 2023; Röösli et al., 2022). Specifically, real-world data often exhibits the long-tail distribution where some demographic groups yield fewer samples than other groups (Cao et al., 2020). Consequently, given the data-driven nature of neural networks, models may be effectively trained in groups with sufficient samples but undertrained in groups only with limited samples, hence resulting in performance disparity across different groups and lower performance for minority groups. On the other hand, recent work suggests that Type I Bias can manifest even when the training set is balanced across

demographic groups (Wang & Deng, 2020). This challenges the conventional understanding of the causes of Type I Bias but promotes the discussion of other possible causes. For instance, Type I Bias may be induced by the underrepresentation of specific demographic groups (Wang & Russakovsky, 2023) or the intrinsic challenges associated with recognizing and classifying specific demographic groups (Klare et al., 2012).

### 7.1.2 Debiasing Methods

Addressing Type I Bias essentially involves optimizing the model to enhance its performance for minority groups while maintaining its performance for majority groups. The strategies can be broadly classified into three main categories based on the stage when the debiasing intervention is applied relative to the model training phase: pre-processing, in-processing, and post-processing. First, pre-processing methods intervene before the training phase. They are primarily designed based on the cause of Type I Bias (the imbalanced distribution across demographic groups in the training set). For instance, the straightforward approach is to construct a balanced real dataset for training (Karkkainen & Joo, 2021) or supplement minority groups with sufficient synthetic training samples (Li & Abd-Almageed, 2023). Another approach in this category involves strategically resampling to increase the occurrence of samples from minority groups or reweighting to assign higher importance to samples from underrepresented groups (Wang & Deng, 2020). Second, in-processing methods are integrated during the model training phase. Most notably, domain adaptation techniques (Kan et al., 2015; Guo et al., 2020) adapt well-learned representations from the majority group to the minority group, and attribute removal methods leverage adversarial learning (Gong et al., 2020; Dhar et al., 2021) to remove demographic information from learned representations. Lastly, post-processing methods apply debiasing techniques after the training process. One common technique is to calibrate the model predictions, ensuring that they adhere to specific fairness criteria (*e.g.*, equalized odds) (Pleiss et al., 2017).

### 7.1.3 Evaluation Protocol

The effectiveness of methods addressing Type I Bias is evaluated by performance disparity between majority and minority groups. In the case of binary attributes, the disparity is directly gauged by the performance difference between majority and minority groups (Buolamwini & Gebru, 2018; Ramaswamy et al., 2021; Xu et al., 2021c). In the case of non-binary attributes, the disparity is gauged by the standard deviation of performance across all demographic groups (STD) (Amini et al., 2019; Gong et al., 2020; 2021; Qin, 2020). To assess performance, there are a variety of metrics such as error rate (Buolamwini & Gebru, 2018; Sattigeri et al., 2019), loss (Hashimoto et al., 2018), accuracy (Kim et al., 2019b), average precision (AP) (Ramaswamy et al., 2021), positive predictive value (PPV), true positive rate (TPR) (Dhar et al., 2021; Adeli et al., 2021), false positive rate (FPR) (Xu et al., 2021c), average false rate (AFR), mean AFR (M AFR) (Ryu et al., 2017), confusion matrix (Gong et al., 2020), F1 score (Adeli et al., 2021), receiver operating characteristic curve (ROC) (Wang & Deng, 2020; Sattigeri et al., 2019; Mirjalili et al., 2018; Qin, 2020; Yu et al., 2020), area under the ROC (AUC) (Mirjalili et al., 2019; Gong et al., 2020; Adeli et al., 2021). Furthermore, besides these metrics to assess performance disparity, the performance improvement in minority groups compared to the baseline is provided for an intuition of debiasing effectiveness, along with overall performance to illustrate that it is not compromised.

### 7.1.4 Datasets

Datasets used to investigate Type I Bias mainly exhibit long-tail distributions. Most notably, several benchmark biometric datasets including LFW (Huang et al., 2007), IJB-A (Klare et al., 2015), IJB-C (Maze et al., 2018), and RFW (Wang et al., 2019b), are frequently utilized. A comprehensive list of datasets is presented in Tab. 13.

### 7.1.5 Future Directions

While many existing works address Type I Bias through balanced sampling or reweighting, more effort should be directed toward designing benchmarks that assess generalization across underrepresented demographic groups, especially in the long-tail regime. Another promising future direction is to delve into the root cause of Type I Bias since the formerly widely accepted cause (data imbalance) has been challenged by the experiment

Table 13: The well-known datasets used to study Type I Bias.

| Name | # Subjects | # Samples | Sex (%) | | Race (%) | | | | |
| --- | --- | --- | --- | --- | --- | --- | --- | --- | --- |
| | | | Female | Male | European | Asian | Indian | African | Hispanic or Latino |
| CelebA Liu et al. (2015) | 10K | 202.5K | 58.3 | 41.7 | - | - | - | - | - |
| MUCT Milborrow et al. (2010) | 0.2K | 3.7K | 50.9 | 49.1 | - | - | - | - | - |
| RaFD Langner et al. (2010) | 67 | 1.6K | 37.3 | 62.7 | - | - | - | - | - |
| PPB Buolamwini & Gebru (2018) | 1.2K | 1.2K | 44.6 | 55.4 | 48.0 | - | - | 52.0 | - |
| MORPH Ricanek & Tesafaye (2006) | 13.6K | 55.1K | 15.3 | 84.7 | 19.2 | 0.28 | - | 77.2 | 3.2 |
| LFW Huang et al. (2007) | 5.7K | 13K | 22.3 | 77.6 | 69.9 | 13.2 | 2.9 | 14.0 | - |
| CASIA-Webface Yi et al. (2014) | 10K | 0.5M | 58.9 | 41.1 | 84.5 | 2.6 | 1.6 | 11.3 | - |
| VGGFace2 Cao et al. (2018) | 8.6K | 3.1M | 59.3 | 40.7 | 74.2 | 6.0 | 4.0 | 15.8 | - |
| MS-Celeb-1M Guo et al. (2016) | 90K | 5.0M | - | - | 76.3 | 6.6 | 2.6 | 14.5 | - |
| IJB-A Maze et al. (2018) | 0.5K | 5.7K | - | - | 66.0 | 9.8 | 7.2 | 17.0 | - |
| IMDB-WIKI Rothe et al. (2018) | 20K | 500K | 41.1 | 57.1 | 79.5 | 2.6 | 2.3 | 11.5 | 4.1 |
| UTK Zhang et al. (2017) | - | 20K | Balanced | | 45.3 | 14.7 | 18.4 | 21.6 | - |
| RFW Wang et al. (2019b) | 12K | 40K | 27.7 | 72.3 | Balanced | | | | - |
| FairFace Karkkainen & Joo (2021) | - | 108K | Balanced | | Balanced | | | | |

that Type I Bias exists even for a balanced dataset (Wang & Deng, 2020). Furthermore, exploring more effective debiasing methods to achieve even performance across cohorts is always of significant importance, hence, it is a valuable direction.

## 7.2 Type II Bias

### 7.2.1 Underlying Causes

The association between prediction targets and attributes in the training set is widely considered the possible cause of Type II Bias (Nam et al., 2020; Zhu et al., 2021; Zhao et al., 2023a). Different from Type I Bias, which originates from an uneven distribution of samples across attributes, Type II Bias arises from an uneven distribution of *specific target groups* across attributes. Specifically, the collected data may encompass a greater number of samples annotated with specific pairs of target labels and attribute labels (*e.g.*, $(y^1, a^1)$ and $(y^2, a^2)$) than other combinations. Models trained on this dataset may leverage these attributes as shortcut features to simplify the training process rather than acquiring more comprehensive features. Consequently, when applying the trained models in real-world scenarios where the association does not generally exist, they may still rely on these attributes for decision-making and yield predictions that depend on these attributes, thereby resulting in a higher frequency of particular prediction outcomes for particular groups and further unfair treatment for these groups.

### 7.2.2 Debiasing Methods

Addressing Type II Bias essentially involves acquiring representations that are independent of the attribute while remaining informative for a wide range of downstream tasks (Balunovic et al., 2022). Similar to Type I Bias, the strategies can be classified into three categories: pre-processing, in-processing, and post-processing. First, pre-processing approaches can be further sub-categorized into dataset construction and data preprocessing. Dataset construction mainly encompasses collecting large-scale universal datasets to lessen the likelihood of spurious correlation between the target and the attribute (Li et al., 2023a;b), and generating counterfactual synthetic samples to augment the original biased training set, thereby reducing its inherent bias strength (Sauer & Geiger, 2021; Kim et al., 2021; Goel et al., 2021; Ramaswamy et al., 2021). Data preprocessing mainly encompasses fairness through unawareness (Dwork et al., 2012), which directly eliminates attributes from the input data, and domain randomization (Tobin et al., 2017) to utilize domain knowledge to assign a random value to the attribute label for each sample, thereby rendering it irrelevant to the target prediction. Second, in-processing approaches can be further divided into two subgroups: methods that either explicitly or implicitly minimize the mutual information (MI) between the learned latent features and the specific attribute. Specifically, several methods directly minimize mutual information between the latent representation for the target classification and the protected attributes to learn a representation that is predictive of the target but independent of the attributes (Kim et al., 2019a; Ragonesi et al., 2021; Zhu

et al., 2021). Another group of methods applies adversarial learning with surrogate losses (Nam et al., 2020; Alvi et al., 2018; Zhang et al., 2018) to implicitly reduce the mutual information or utilize domain-invariant learning (Ganin et al., 2016; Zhao et al., 2019b; Albuquerque et al., 2019; Sagawa* et al., 2020; Ahmed et al., 2021; Creager et al., 2021) to minimize classification performance gap across groups by mapping data to a space where distributions are indistinguishable while maintaining task-relevant information. Lastly, for the post-processing method, domain-independent learning (Wang et al., 2020b) learns an ensemble classifier comprising separate classifiers for each demographic group by sharing representations, thereby ensuring that the prediction from the unified model is not biased towards any domain.

### 7.2.3 Evaluation Protocol

The effectiveness of methods addressing Type II Bias is evaluated by prediction disparity across different groups. In the prevalent evaluation protocol, models are trained on a dataset where the target is associated with the attribute and tested on a held-out dataset where such association is absent (Kim et al., 2019a; Ragonesi et al., 2021; Zhu et al., 2021). Subsequently, the testing accuracy is reported to evaluate the model's capability to reduce the effect of association in the training set (the effectiveness to mitigate Type II Bias) (Wang et al., 2020b). Several studies also present the accuracy of worst-case groups, where the samples yield the opposite of the bias present in the training set (Sagawa* et al., 2020; Liu et al., 2021a; Lee et al., 2021). Furthermore, we summarize other commonly-used bias assessment metrics in Tab. 12. A noteworthy distinction in these bias assessment metrics for Type II Bias compared with Type I Bias is the absence of necessity for ground truth labels. This distinction is attributed to the fact that Type II Bias is defined as the dependence between model prediction and attribute, eliminating the need for ground truth, while evaluating Type I Bias necessitates ground truth to assess model performance.

### 7.2.4 Datasets

Most notably, several census datasets, including the Adult income dataset (Dua & Graff, 2017), German credit dataset (Dua & Graff, 2017), and COMPAS recidivism dataset (Angwin et al., 2022a), are employed as benchmark datasets to investigate the impact of sensitive/protected attributes in real-world decision-making processes. Additionally, computer vision and natural language processing communities also develop various datasets to investigate Type II Bias, *e.g.*, Colored MNIST (Kim et al., 2019a), CelebA (Liu et al., 2015; Nam et al., 2020), Waterbirds (Sagawa* et al., 2020), and CivilComments-WILDS (Borkan et al., 2019; Koh et al., 2021). A comprehensive list of datasets is summarized in Tab. 14.

Table 14: The well-known datasets used to study Type II Bias.

| Name | Modality | Attribute | Target |
|---|---|---|---|
| Adult Dua & Graff (2017) | Tabular | Sex | Income |
| German Dua & Graff (2017) | Tabular | Sex, age | Credit |
| COMPAS Angwin et al. (2022a) | Tabular | Race | Recidivism |
| Colored MNIST Kim et al. (2019a) | Image | Color | Digit |
| CelebA Liu et al. (2015) | Image | Sex | Facial attributes |
| IMDB Rothe et al. (2015) | Image | Sex, age | Age, sex |
| Waterbirds Sagawa* et al. (2020) | Image | Background | Waterbirds or landbirds |
| CivilComment-WILDS Koh et al. (2021) | Text | Demographic identities | Toxic or non-toxic |
| WinoBias Zhao et al. (2018) | Text | Gender pronouns | Coreference resolution |
| Bias in Bios De-Arteaga et al. (2019a) | Text | Gender | Occupation |
| MS-COCO Lin et al. (2014) | Text & Image | Gender | Object |

### 7.2.5 Future Directions

Type II Bias highlights the importance of learning representations that are causally disentangled from protected or spurious attributes. Future work should focus on developing provably invariant representations using tools from causality, domain invariance, and contrastive learning. Another promising research direction

is to explore the strong bias region (Li et al., 2023a) of Type II Bias, where the target and the attribute are strongly associated in the training set, a scenario that is overlooked by many existing works (Alvi et al., 2018; Kim et al., 2019a). Also, it is important to further explore more challenging scenarios where attribute labels are absent (Wang et al., 2019a; Bahng et al., 2020; Cadene et al., 2019) or unknown biases emerge (Li & Xu, 2021; Zhang et al., 2022b; Creager et al., 2021). Future research should study semi-supervised or unsupervised debiasing methods that can discover hidden biases and adapt accordingly, without relying on full attribute supervision. Moreover, a key challenge in Type II Bias is the lack of reliable diagnostic tools to detect when a model relies on protected attributes. New metrics or interpretability techniques are needed to identify attribute leakage, especially in high-dimensional representations.

### 7.3 Summary

In this section, we highlight the distinctions between Type I Bias and Type II Bias across multiple aspects and provide further explanations on the comparison in Tab. 1.

- Manifestation. A model exhibiting Type I Bias yields uneven performance across different groups and lower performance in minority groups, whereas a model exhibiting Type II Bias depends on attributes for decision-making and produces specific predictions that are highly associated with specific attributes.

- Disparity. Type I Bias refers to the disparity in prediction performance across attributes, whereas Type II Bias refers to the disparity in prediction outcomes across attributes.

- Causes. Type I Bias stems from insufficient training of underrepresented groups, whereas Type II Bias arises from the association between targets and attributes.

- Dataset inducing bias. An imbalanced distribution of samples across attributes induces Type I Bias, whereas an imbalanced distribution of *specific target groups* across attributes induces Type II Bias.

## 8 Suggestions

In this section, we first introduce a general framework for distinguishing between Type I and Type II Bias. We then offer practical guidance tailored to different audiences to help prevent confusion between the two types.

### 8.1 Guiding Framework for Categorizing Biases

To promote clarity in practical applications, we propose the following decision flow for identifying the bias type based on observable phenomena and context:

- **Performance disparities across demographic groups:** If lower accuracy or higher error rates are observed for certain groups (*e.g.*, minority groups), this may correspond to **Type I Bias**.

- **Systematic changes in model output with respect to protected attributes:** If the model's predictions change systematically as protected attributes vary, this may indicate **Type II Bias**.

When still in doubt, consider the used evaluation metric:

- *Does the evaluation require ground truth labels?*
    - Yes → Type I Bias (*e.g.*, fairness metrics such as True Positive Rate (TPR), False Positive Rate (FPR), or accuracy).
    - No → Type II Bias (*e.g.*, evaluating mutual information between the model prediction and a protected attribute).

### 8.2 Audience-Specific Guidance

In this section, we provide some suggestions for different audiences to help them identify and distinguish between Type I Bias and Type II Bias.

- Researchers:
  - Explicitly and precisely specify the type of bias being addressed to avoid vague terminology. Using well-defined terms such as Type I Bias and Type II Bias ensures clarity.
  - Derive motivation from prior work that addresses the same type of bias. This alignment helps reduce existing confusion in the literature.
  - Avoid introducing new terminology for previously studied biases. If a new term is necessary, clearly distinguish it from existing definitions. Reusing established terms fosters a more unified and comprehensible research community.

- ML Practitioners: When deploying models, evaluate Type I Bias through accuracy disparity and related performance metrics. Assess Type II Bias using measures of attribute dependence such as mutual information analysis.

- Educators: Illustrate the two bias types using concrete examples (*e.g.*, face recognition for Type I Bias and credit lending for Type II Bias). Use guiding questions such as:
  - "Is the model performing worse for Group A?"    Yes $\rightarrow$ Type I Bias
  - "Would changing the attribute label (*e.g.*, sex) change the model's prediction?"    Yes $\rightarrow$ Type II Bias

## 9 Conclusion

Through an investigation of 415 papers, we uncover the substantial confusion surrounding two prevalent types of biases within the machine learning community, which amplifies the learning burden for new researchers. Subsequently, we delve into the possible causes of the confusion. Most notably, we observe that researchers from diverse backgrounds hold different preconceptions about bias, leading to a lack of unified terminology for the same type of bias over an extended period. To alleviate the existing confusion and restore clarity in the literature, we present mathematical definitions for these two prevalent types of biases. Furthermore, we unify a comprehensive list of papers under these definitions and distinguish these two types of biases from multiple perspectives. Through this endeavor, we seek to facilitate the discussion on bias-related issues among researchers with diverse backgrounds.

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
