# Appendix

## 1 Full Categorization

In this section, we provide a comprehensive list of all 415 papers that investigate Type I Bias and Type II Bias. Besides, for more fine-grained categorization, we classify papers addressing predominant issues into various subgroups.

### 1.1 Type I Bias

#### 1.1.1 Biometrics

Wang & Deng (2020); Wang et al. (2019b); Gong et al. (2021); Liu et al. (2022a); Xu et al. (2021c); Salvador et al. (2022); Qin (2020); Ryu et al. (2017); Yucer et al. (2020); Yu et al. (2020); McDuff et al. (2019); Long et al. (2015); Liu et al. (2021b); Dhar et al. (2020); Terhörst et al. (2020a); Terhörst et al. (2020b); Robinson et al. (2020); Serna et al. (2022); Li & Abd-Almageed (2023); Pahl et al. (2022); Conti & Clémençon (2024).

Investigation of the Role of Demographic Information: Klare et al. (2012); Dhar et al. (2021); Gong et al. (2020); Morales et al. (2020); Suriyakumar et al. (2023); Kang et al. (2020); Albiero & Bowyer (2020); Cavazos et al. (2020); Han et al. (2017); Nagpal et al. (2019); Grother et al. (2019).

#### 1.1.2 Classification of Protected Attribute

Buolamwini & Gebru (2018); Balakrishnan et al. (2021); Karkkainen & Joo (2021); Das et al. (2018); Weerts et al. (2023); De Vries et al. (2019); Cruz & Hardt (2024).

#### 1.1.3 Other Tasks Associated with Protected Attribute

Hashimoto et al. (2018); Xu et al. (2021b); Shen et al. (2023); Taori & Hashimoto (2023); Vu et al. (2022); Wan (2022); Cheng et al. (2021); Tan & Celis (2019); Han et al. (2023); Chai & Wang (2022b); Ji et al. (2020); Rahmattalabi et al. (2019); Samadi et al. (2018); Yurochkin et al. (2019); Kim & Cho (2020); Shankar et al. (2017); Li et al. (2014); Xiao et al. (2023); Bell & Sagun (2023); Shrestha et al. (2023); Mccradden et al. (2023); Gardner et al. (2023); Hutiri & Ding (2022); Pastaltzidis et al. (2022); Markl (2022); Donahue et al. (2022); Wang et al. (2022a); Fong et al. (2021); McLaughlin et al. (2022); Steed & Caliskan (2021); Dhamala et al. (2021); Martin & Wright (2023); Nanda et al. (2021); Sweeney & Najafian (2020); Yang et al. (2020b); Green & Chen (2019); Belém et al. (2024).

#### 1.1.4 Equalized Odds

Park et al. (2022); Zhang et al. (2023); Conti et al. (2022); An et al. (2022); Schrouff et al. (2022); Konstantinov & Lampert (2022); Gölz et al. (2019); Kallus & Zhou (2018); Pleiss et al. (2017); Kim & Cho (2020); Quadrianto & Sharmanska (2017); Loi & Heitz (2022); Blum et al. (2022); Nandy et al. (2022); Wu & He (2022); Mishler et al. (2021); Wang et al. (2021); Taskesen et al. (2021); Coston et al. (2020); Canetti et al. (2019); Chang et al. (2023); Plecko & Bareinboim (2024); Cherian & Candès (2024); Small et al. (2024); Simson et al. (2024).

#### 1.1.5 Equal Opportunity

Quadrianto et al. (2019); Jung et al. (2022); Yu et al. (2022); Pham et al. (2023); Creager et al. (2020); Dutta et al. (2020); Sabato & Yom-Tov (2020); Donini et al. (2018); De-Arteaga et al. (2019b); Henzinger et al. (2023); Awasthi et al. (2021); D'Amour et al. (2020); Hu & Chen (2020); Heidari et al. (2019); Kearns et al. (2019); Corbett-Davies et al. (2017); Tang et al. (2024); Selialia et al. (2024); Binkyte et al. (2024).

### 1.1.6 Accuracy Parity

Kim et al. (2019b); Quan et al. (2023); Zafar et al. (2017a); Zhao et al. (2019a); Zhang & Long (2021); Xu et al. (2021a); Mickel (2024); Baumann et al. (2024); Akpinar et al. (2024); Lünich & Keller (2024); Dong et al. (2024); Lee et al. (2024).

## 1.2 Type II Bias

### 1.2.1 Labeled Spurious Attribute

Sagawa* et al. (2020); Rosenfeld et al. (2022); Kirichenko et al. (2023); Izmailov et al. (2022); Levy et al. (2020); Zhu et al. (2021); Ragonesi et al. (2021); Kim et al. (2019a); Alvi et al. (2018); Tartaglione et al. (2021); Hong & Yang (2021); Hwang et al. (2022); Bevan & Atapour-Abarghouei (2022); Chen & Joo (2021); Wang et al. (2020b); Ramaswamy et al. (2021); Li et al. (2018); Li & Vasconcelos (2019); Fan et al. (2022); Chu et al. (2021); Wen et al. (2021); Mo et al. (2021); Cadene et al. (2019); Idrissi et al. (2022); Clark et al. (2019a); Goel et al. (2021); Sauer & Geiger (2021); Adila et al. (2024); Albuquerque et al. (2024); Jung et al. (2024); Zeng et al. (2024); Sreelatha et al. (2024); Chakraborty et al. (2024); Alabdulmohsin et al. (2024).

Simplicity Bias: Xue et al. (2023); Tiwari & Shenoy (2023); Addepalli et al. (2023); Trivedi et al. (2023); Lyu et al. (2021); Gatmiry et al. (2024); Tsoy & Konstantinov (2024); Rende et al. (2024); Nguyen et al. (2019); He et al. (2024); Chen et al. (2024a).

Shape and Texture Bias: Mummadi et al. (2021); Li et al. (2021b); Gat et al. (2020); Mishra et al. (2020); Singh et al. (2020).

### 1.2.2 Unlabeled Spurious Attribute

Geirhos et al. (2018); Wang et al. (2019a) Bahng et al. (2020); Clark et al. (2019b); He et al. (2019); Cadene et al. (2019); Clark et al. (2020); Utama et al. (2020).

### 1.2.3 Unknown Spurious Attribute

Zhao et al. (2023a); Li & Xu (2021); Li et al. (2022); Jeon et al. (2022); Le Bras et al. (2020); Kim et al. (2021); Du et al. (2021); Yaghoobzadeh et al. (2019); Sanh et al. (2021); Lahoti et al. (2020); Pezeshki et al. (2021); Nam et al. (2020); Lee et al. (2021); Liu et al. (2021a); Liu et al. (2023); Creager et al. (2021); Zhang et al. (2022b); Nam et al. (2022); Ahmed et al. (2021); Taghanaki et al. (2021); Sohoni et al. (2020); Ahn et al. (2023); Jung et al. (2023b); Kim et al. (2022); Qiu et al. (2023); Barbano et al. (2023); Basu et al..

### 1.2.4 Labeled Sensitive Attribute

Calders & Verwer (2010); Angwin et al. (2022b); Xu et al. (2022b); Jesus et al. (2022); Yang et al. (2022); Zong et al. (2023); Kong et al. (2021); Tucker & Shah (2022); Liang et al. (2021); Vig et al. (2020); Celis et al. (2020); Bolukbasi et al. (2016); Sadeghi et al. (2019); Grari et al. (2021); Sun et al. (2019); Mei et al. (2023); Wolfe et al. (2023); Cabello et al. (2023); Bianchi et al. (2023); Hirota et al. (2022); Ball-Burack et al. (2021); Cho et al. (2021); Obermeyer et al. (2019); Jung et al. (2025a); Jung et al. (2025b); Teo et al. (2024); Kim et al. (2024); Limisiewicz et al. (2024); Shen et al. (2024); Dehdashtian et al. (2024b).

### 1.2.5 Unknown Sensitive Attribute

Chai & Wang (2022a); Buet-Golfouse & Utyagulov (2022); Lu et al. (2024).

### 1.2.6 Demographic Parity

Creager et al. (2019); Xu et al. (2018); van Breugel et al. (2021); Liu et al. (2022b); Grazzi et al. (2022); Gaucher et al. (2022); Buyl & De Bie (2022); van der Linden et al. (2022); Lohaus et al. (2022); Shahin Shamsabadi et al. (2022); Alabdulmohsin et al. (2022); Xian et al. (2023); Singh et al. (2023); Jovanović et al. (2023); Cruz et al. (2023); Yang et al. (2023b); Giguere et al. (2022); Shui et al. (2022a);

Yan & Zhang (2022); Alabdulmohsin & Lucic (2021); Bello & Honorio (2020); Chzhen et al. (2020); Oneto et al. (2020); Roh et al. (2020); Gordaliza et al. (2019); Zhao & Gordon (2022); Locatello et al. (2019); Cunningham & Delany (2021); Zafar et al. (2017b); Yang et al. (2023a); Chen et al. (2023b); Rateike et al. (2022); Ghazimatin et al. (2022); Zhang & Davidson (2021); Jin et al. (2024a); Xiong et al. (2023); Ohayon et al. (2024); Defrance et al. (2024); Vladimirova et al. (2024); Dehdashtian et al. (2024a); Liu et al. (2024); Kang et al. (2024); Cachel & Rundensteiner (2024); Yeh et al. (2024).

## 1.3 Both Type I and Type II Biases

Amini et al. (2019); Adeli et al. (2021); Stone et al. (2022); Wang & Russakovsky (2023).

### 1.3.1 Fairness Criteria

Shui et al. (2022b); Zhang et al. (2022a); Chai et al. (2022); Hsu et al. (2022); Chen et al. (2022); Mehrabi et al. (2021b); Soen et al. (2022); Alghamdi et al. (2022); Sattigeri et al. (2022); Li et al. (2023c); Zhu et al. (2022); Khalili et al. (2023); Soen et al. (2023); Mangold et al. (2023); Roh et al. (2023); Hosseini et al. (2023); Jung et al. (2023a); Deng et al. (2023); Balunovic et al. (2022); Zhang et al. (2022c); Li & Liu (2022); Wang et al. (2022b); Chai & Wang (2022b); Jin et al. (2022); Du et al. (2021); Roh et al. (2021b); Bendekgey & Sudderth (2021); Ding et al. (2021); Aïvodji et al. (2021); Li et al. (2021a); Chuang & Mroueh (2021); Roh et al. (2021a); Celis et al. (2021); Wang et al. (2020a); Yang et al. (2020a); Mandal et al. (2020); Cho et al. (2020); Savani et al. (2020); Kim et al. (2020); Mozannar et al. (2020); Saha et al. (2020); Lohaus et al. (2020); Zhao et al. (2020); Baharlouei et al. (2020); Williamson & Menon (2019); Chzhen et al. (2019); Lamy et al. (2019); Madras et al. (2018); Liu et al. (2018); Kilbertus et al. (2018); Kearns et al. (2018); Agarwal et al. (2018); Yao & Huang (2017); Xu et al. (2022c); Lokhande et al. (2020); Sattigeri et al. (2019); Beutel et al. (2017); Zhang et al. (2018); Zemel et al. (2013); Xu et al. (2019); Richardson et al. (2023); Bell et al. (2023); Defrance & De Bie (2023); Calvi & Kotzinos (2023); Ganesh et al. (2023); Petersen et al. (2023); Alvarez et al. (2023); Almuzaini et al. (2022); Black et al. (2022); Baumann et al. (2022); Mishler & Kennedy (2021); Grabowicz et al. (2022); Zhang (2022); Kong (2022); Sikdar et al. (2022); Pfohl et al. (2022); Agarwal & Deshpande (2022); Sharaf et al. (2022); Singh et al. (2021); Räz (2021); Rodolfa et al. (2020); Slack et al. (2020); Liu et al. (2020); Harrison et al. (2020); Kallus et al. (2022); Celis et al. (2019); Friedler et al. (2019); Menon & Williamson (2018); Becker et al.; Sharma & Deshpande (2023); Tifrea et al.; Xu et al.; Schrouff et al. (2024); Zhang et al.; Pang et al. (2025); Taufiq et al.; Luo et al. (2024); Wang et al. (2024); Chen et al. (2024c); Grari et al. (2024); Chowdhury et al. (2024); Chen et al. (2024c); Yin et al. (2024); Liu & Zhao (2024); Tian et al. (2024); Weerts et al. (2024); Zezulka & Genin (2024); Poe & El Mestari (2024); Ni et al. (2024); Chan et al. (2024); Somerstep et al. (2024); Laszkiewicz et al. (2024); Gillis et al. (2024); Jaime & Kern (2024); Blandin & Kash (2024); Rateike et al. (2024); Wyllie et al. (2024); Mhasawade et al. (2024).

## 1.4 Survey about Bias Issues

Qian et al. (2021); Chen et al. (2023a); Du et al. (2020); Castelnovo et al. (2022); Wang et al. (2022c); Mehrabi et al. (2021a); Fabbrizzi et al. (2022); Le Quy et al. (2022); Corbett-Davies & Goel (2018); Berk et al. (2021); Blodgett et al. (2020); Benbouzid (2023); Devinney et al. (2022); Schwöbel & Remmers (2022); Finocchiaro et al. (2021); Hutchinson & Mitchell (2019); Binns (2018); Baumann et al. (2023); Bellamy et al. (2019); Hort et al. (2022) Caton & Haas (2020); Delaney et al. (2024); Jin et al. (2024b); Buyl et al. (2024); Han et al. (2024); Deck et al. (2024).

## 1.5 Bias Assessment Metrics

Jiang et al. (2021); Zhao et al. (2017); Wang & Russakovsky (2021); Zhao et al. (2023b); Wang et al. (2019c); Chen & Wu (2020); Li & Abd-Almageed (2021); Zafar et al. (2017a); Morales et al. (2020); Mirjalili et al. (2019); Creager et al. (2019); Székely et al. (2007); Li & Abd-Almageed (2023); Zafar et al. (2017b); Li et al. (2018) Li & Vasconcelos (2019); Xie et al. (2017); Hiranandani et al. (2020); Leino et al. (2018).

### 1.6 Fairness Constraints

Hardt et al. (2016); Calders et al. (2009); Kusner et al. (2017); Dwork et al. (2012); Chen et al. (2019); Calders & Verwer (2010); Lechner et al. (2021); Grgic-Hlaca et al. (2016); Bellamy et al. (2019); Kamiran & Calders (2012); Kang et al. (2022); Bechavod (2024); Munagala & Sankar (2024); Xu et al. (2024).

## 2 Datasets Construction

In this section, we introduce dataset setups designed to isolate the presence of a specific type of bias—either Type I Bias or Type II Bias—while ensuring the other type is absent.

### 2.1 Type I Bias Exists without Type II Bias

We synthesize training set w.r.t. $A, X, Y$ by the following generative model,

$$
\begin{aligned}
&A \sim \mathrm{Ber}(1/100) \times 2 - 1; \\
&V_1 \sim \mathrm{Norm}(-1, \sigma = 0.2); \\
&V_2 \sim \mathrm{Norm}(1, \sigma = 0.2); \\
&T \sim \mathrm{Ber}(1/2); \\
&U|_{A=1} \sim V_1 \times T + V_2 \times (1 - T); \\
&U|_{A=-1} \sim U|_{A=1} - 1; \\
&X = [U, A]^T; \\
&Y \sim \mathbb{1}_{U>0};
\end{aligned}
$$

where $\mathrm{Ber}(p)$ represents the Bernoulli distribution with probability $p$, $\mathrm{Norm}(\mu, \sigma)$ represents the normal distribution with mean $\mu$ and standard deviation $\sigma$, and $\mathbb{1}$ is the indicator function. As shown in Fig. 3, the training set is imbalanced across attribute $A$, with the subset where $A = -1$ being the minority group. Furthermore, the optimal classification boundary is set to be varied across $A$ since one widely accepted cause of Type I Bias is that the model trained on the sufficient samples in majority groups might not effectively generalize to minority groups (Wang & Russakovsky, 2023). Additionally, the testing set is constructed using the following generative model,

$$
\begin{aligned}
&A \sim \mathrm{Ber}(1/2) \times 2 - 1; \\
&V_1 \sim \mathrm{Norm}(-1, \sigma = 0.2); \\
&V_2 \sim \mathrm{Norm}(1, \sigma = 0.2); \\
&T \sim \mathrm{Ber}(1/3); \\
&U|_{A=1} \sim V_1 \times T + V_2 \times (1 - T); \\
&U|_{A=-1} \sim V_1 \times T + V_2 \times (1 - T) - 1; \\
&X = [U, A]^T; \\
&Y \sim \mathbb{1}_{X>0};
\end{aligned}
$$

where $A$ is assigned either value 0 or 1 with equal probability. Hence, the testing set is balanced across values of the attribute.

### 2.2 Type II Bias Exists without Type I Bias

In this section, we introduce dataset setups designed to isolate the presence of a specific type of bias—either Type I or Type II—while ensuring the other type is absent.

We synthesize training set w.r.t. $A, X, Y$ by the following generative model,

$$A \sim \text{Ber}(1/2) \times 2 - 1;$$
$$V_1 \sim \text{Norm}(-1, \sigma = 0.2);$$
$$V_2 \sim \text{Norm}(1, \sigma = 0.2);$$
$$T \sim \text{Ber}(1/100);$$
$$U|_{A=1} \sim V_1 \times (1 - T) + V_2 \times T;$$
$$U|_{A=-1} \sim V_1 \times T + V_2 \times (1 - T);$$
$$X = [U, A]^T;$$
$$Y \sim \mathbb{1}_{X>0}.$$

As shown in Fig. 4, the training set yields more samples with combinations $A = 1, Y = 0$ and $A = -1, Y = 1$ compared to other combinations. This setting is motivated by that the association between target $Y$ and attribute $A$ in the training set is considered one widely-accepted reason for Type II Bias (Nam et al., 2020; Zhu et al., 2021; Tartaglione et al., 2021). The testing set is generated to be balanced across both $Y$ and $A$ with the following generative model,

$$A \sim \text{Ber}(1/2) \times 2 - 1;$$
$$V_1 \sim \text{Norm}(-1, \sigma = 0.2);$$
$$V_2 \sim \text{Norm}(1, \sigma = 0.2);$$
$$T \sim \text{Ber}(1/2);$$
$$U|_{A=1} \sim V_1 \times (1 - T) + V_2 \times T;$$
$$U|_{A=-1} \sim V_1 \times T + V_2 \times (1 - T);$$
$$X = [U, A]^T;$$
$$Y \sim \mathbb{1}_{X>0}.$$