# OpenReview forum: "Fairness and Disentanglement: A Critical Review of Predominant Bias in Neural Networks"
_TMLR — Accepted by TMLR_

### Review · Reviewer_B3WE · 2025-05-03

**Summary Of Contributions:**

This paper proposes two main forms of bias in neural networks: unequal performance between demographic groups (Type I) and reliance on protected attributes for prediction (Type II). After reviewing 415 articles, the authors detect widespread confusion in the literature about these types of biases. To address it, they propose mathematical formulations for both, unify previous work under these definitions, and conduct experiments with synthetic and real data to demonstrate that these biases are distinct, with different causes and approaches.

**Audience:**

Yes

**Claims And Evidence:**

Yes

**Requested Changes:**

Weaknesses and improvements needed:

Section 1:
- The first part presents an excessive number of references very close together, which makes the text difficult to read from the beginning. Several references are mentioned without explaining their contribution and some are repeated, which generates confusion. To improve, it is recommended to group them by topic, keep only the most relevant ones and better develop those that exemplify the problem discussed (such as facial recognition systems or bank loans).
- Table 1 includes technical notation that has not been presented previously. If it is left in this section, it should either be explained in non-mathematical language or moved to the next section.
- There is no mention of Figure 1 in this part of the text. Although its purpose seems to be to illustrate the conceptual confusion about bias in the literature,  it is not connected to the text explicitly.
- In the sentence “the taxonomy of bias issues in existing survey papers may not sufficiently distinguish...”, it is not clear what omissions are detected or how they contribute to the confusion.
- In general, the idea of “confusion” is repeated in several parts without clear examples or developments, with repetitive vocabulary and vague expressions.

Sections 2 and 4:
- The notation “D : X, Y, A” is ambiguous. It would be better to define D as a set of instances (x,y,a) and then describe the domains of these variables.
- Sections 2.1 and 2.2 also present an overabundance of citations that could be reduced or restructured.
- Section 4.1 reintroduces metrics whose names and abbreviations had already been introduced in section 2.1.

Sections 5 and 6:
- It is suggested to evaluate the possibility of integrating both sections into a single thematic block that addresses the types of confusion observed, their causes and implications, for greater clarity and to avoid excessive fragmentation.

Sections 7 and 9:
- Section 7 includes many technical details that make it difficult to read and follow. Consider reorganizing it to highlight the main findings and moving the more technical aspects to the appendix.
- In subsection 7.3, consider including a table with statistics from the CelebA dataset (as is done in Table 9 for the Adult dataset) to contribute to the consistency of the analysis.
- Section 9 is key to the conceptual closure of the article. It would be useful to provide a more practical guiding framework for different audiences than researchers to help them properly categorize biases or choose appropriate metrics.

Questions for discussion and clarification:
- The current definitions of Type I and Type II bias treat any observed difference, regardless of its magnitude, as indicative of bias. While this may hold conceptually, what happens in practice? For example, a 1% disparity might be critical in healthcare but trivial in marketing.
- How does this formulation generalize to multi-class settings or to tasks beyond classification?
- What guidance is offered for evaluating systems or metrics that aim to address both types of bias simultaneously?
- Are there cases where confusion between these types of bias has resulted in incorrect conclusions or decisions?
- How do other taxonomies differ compared to the proposed one?

General considerations:
- There are inconsistencies in citation style throughout the text. It is recommended that all citations at the end of a sentence be enclosed in parentheses.
- Improve descriptions of tables and figures. Ensure that the font size in tables is legible and consistent (e.g., Table 3 is difficult to read).
- The manuscript is quite long and several sections are very fragmented. It is recommended to improve the overall organization and move more detailed content to the appendix to make it easier to follow and read.

**Strengths And Weaknesses:**

Strengths:
- The paper provides formal mathematical definitions for two types of bias, which offers a useful framework for the community.
- The exhaustive review of 415 papers following a clear methodology and their classification according to the type of bias they address is novel.
- Experiments on synthetic and real data provide empirical validation of the distinction between the two types of bias.

Weaknesses:
- See Requested Changes.

---

> ### Author Response · Authors · 2025-05-24
>
> Thank you for your detailed suggestion. We have revised the paper accordingly and highlighted the changes in orange.
>
> **The idea of “confusion”.**
>
> As illustrated in Figure 1, this confusion arises when researchers encounter multiple works that all reference ``bias" without clearly specifying its nature. We modified the paper accordingly.
>
>
> **Q1.**
>
> In the general case, our definitions treat any non-zero disparity as bias (Definitions 1 and 2) to ensure conceptual clarity. This approach reflects the practical nature of bias definitions, which are more applicable in real-world scenarios where bias is frequently encountered, unlike fairness, often seen as an ideal goal that is rarely achieved in practice, as discussed in Section 4.4. Besides, we consider the magnitude of disparity as a reflection of bias strength, which we empirically investigate in Section 6.2 using real-world datasets. In practice, we acknowledge that the significance of a given disparity is inherently context-dependent. To address this, we revised the manuscript to clarify that applying domain-specific thresholds when interpreting bias metrics is advisable in practical deployments.
>
>
> **Q2.**
>
> Our definitions can be applied beyond binary classification. For multi-class settings, the same principles hold: Type I Bias manifests as performance disparity across attribute groups, and Type II Bias as dependence between predictions and attributes. For regression tasks, Type I Bias can be assessed using performance metrics such as mean squared error (MSE), and Type II Bias can be evaluated using standard measures of attribute dependence such as mutual information. We elaborate on this extension in Section 7.
>
> **Q3.**
>
> In Section 6.3, we evaluate metrics tailored to each bias type and compare their effectiveness. For systems addressing both biases, we recommend evaluation using separate metrics for Type I Bias (e.g., accuracy disparity, equal opportunity) and Type II Bias (e.g., demographic parity, mutual information). We also highlight that certain methods (e.g., invariant representation learning) can impact both biases but should still be assessed distinctly.
>
>
> **Q4.**
>
> Yes. Section 5 provides several concrete examples. For instance, Ragonesi et al. (2021) [1] and Alvi et al. (2018) [2] address Type II Bias but motivate their work using uneven performance, a Type I Bias manifestation. Conversely, Wang & Deng (2021) [3] address Type I Bias but are motivated by literature on prediction dependence. This mismatch leads to evaluation with inappropriate metrics, which can misrepresent method effectiveness. We believe such confusion has hindered fair comparisons and the development of targeted debiasing strategies.
>
> [1] Ruggero Ragonesi, Riccardo Volpi, Jacopo Cavazza, and Vittorio Murino. Learning unbiased representations via mutual information backpropagation. In Proceedings of the IEEE/CVF Conference on Computer Vision and Pattern Recognition, pp. 2729–2738, 2021.
>
> [2] Mohsan Alvi, Andrew Zisserman, and Christoffer Nellåker. Turning a blind eye: Explicit removal of biases and variation from deep neural network embeddings. In Proceedings of the European Conference on Computer Vision (ECCV) Workshops, pp. 0–0, 2018.
>
> [3] Mei Wang and Weihong Deng. Deep face recognition: A survey. Neurocomputing, 429:215–244, 2021.
>
>
> **Q5.**
>
> Existing taxonomies (e.g., Mehrabi et al., 2021 [4]; Wang et al., 2019b [5]) categorize biases by source (e.g., dataset, algorithm) or domain (e.g., racial, gender). However, as illustrated in Table 6, the same terminology, such as "gender bias," is often used to refer to fundamentally different underlying issues depending on context, which undermines the clarity and significance of reported findings. In contrast, our taxonomy is different in formally defining the two predominant manifestations—performance disparity (Type I Bias) and prediction dependence (Type II Bias)—and using these to unify prior work (Sec. 4). This manifestation-based approach enables precise evaluation and categorization, as evidenced in our unification of 415 papers and alignment with existing fairness criteria.
>
> [4] Ninareh Mehrabi, Fred Morstatter, Nripsuta Saxena, Kristina Lerman, and Aram Galstyan. A survey on bias and fairness in machine learning. ACM Computing Surveys (CSUR), 54(6):1–35, 2021a.
>
> [5] Mei Wang, Weihong Deng, Jiani Hu, Xunqiang Tao, and Yaohai Huang. Racial faces in the wild: Reducing racial bias by information maximization adaptation network. In Proc. of the IEEE/CVF International Conf. on Computer Vision, pp. 692–702, 2019b.

---

### Review · Reviewer_HUqC · 2025-05-05

**Summary Of Contributions:**

_Caveat: I am not a specialist in the fairness domain, and although I am familiar with some of the 415 papers covered in this review, I certainly do not know all of them._

This paper appears to provide a comprehensive survey of the literature that the machine-learning community currently discusses under the banners of fairness and trustworthy AI. In particular, it defines Type I and Type II biases and catalogues how earlier works have addressed them. Along the way, it also summarizes the confusions that have surfaced in the literature, such as the misuse of terminology. Consequently, this work (1) urges researchers to specify explicitly which bias type they are tackling and to adopt unambiguous terms such as "Type I" and "Type II," (2) recommends grounding new studies in prior work that targets the same bias type so the community’s motivation remains coherent, and (3) discourages coining new terms for already-known phenomena unless the difference is meticulously spelled out, thereby fostering a clearer and more unified research landscape.

**Audience:**

Yes

**Claims And Evidence:**

Yes

**Requested Changes:**

Building on the comment about text, I think it would be very helpful---though not strictly necessary---if the paper added some comments that applies its viewpoint to language tasks. Bias in images feels quite concrete (Figure 1 makes this especially clear, perhaps because I am more familiar with fairness work in vision), but I am not sure what the analogous manifestations of Type I and Type II bias look like for text.

**Strengths And Weaknesses:**

Strengths:
- By systematically organizing prior work through the lens of Type I and Type II biases, the paper delivers a comprehensive overview that researchers can adopt as an white paper.
- I personally like Section 5; it directly tackles the terminological confusion that so often clouds fairness research. In my experience, ambiguous wording appears regularly in this field, so spotlighting those pitfalls should help future studies articulate their contributions with far greater precision.

Weaknesses:
- The current spotlight in AI is undeniably on text, yet the paper devotes relatively little space to it; for instance, Table 13 lists three tabular datasets and four image datasets, but only one text dataset. Admittedly, given that this is a review grounded in existing work, this imbalance is understandable and not a major flaw.

---

> ### Author Response · Authors · 2025-05-24
>
> **W1.**
>
> We have revised Table 14 to include additional datasets from the NLP community.
>
>
> **C1.**
>
> While our experimental validation focuses on image and tabular data, the proposed taxonomy and definitions can extend naturally to language tasks. For example:
> - Type I Bias in text classification can manifest as uneven performance across demographic subgroups, such as sentiment classifiers that perform better on reviews written in Standard American English than on African American Vernacular English, even when ground truth labels are consistent [1].
> - Type II Bias in text tasks often arises when model predictions (e.g., toxicity, hate speech, or occupation classification) are influenced by identity-indicative features such as names, pronouns, or phrases associated with a protected group—even when these features are irrelevant to the task [2].
> We include these clarifications to illustrate that our bias definitions apply beyond vision and tabular data, and are also capable of guiding analyses in NLP fairness research.
>
> [1] Rachael Tatman. Gender and dialect bias in youtube’s automatic captions. In Proceedings of the first ACL workshop on ethics in natural language processing, pp. 53–59, 2017.
>
> [2] Isabel O Gallegos, Ryan A Rossi, Joe Barrow, Md Mehrab Tanjim, Sungchul Kim, Franck Dernoncourt, Tong Yu, Ruiyi Zhang, and Nesreen K Ahmed. Bias and fairness in large language models: A survey. Computational Linguistics, 50(3):1097–1179, 2024.

---

### Review · Reviewer_rLRv · 2025-05-10

**Summary Of Contributions:**

The paper presents a critical discussion of how the term *bias* is used the AI literature. It highlights two different types of bias, namely uneven performance across demographic groups (Type I) and decision-making processes relying on protected attributes (Type II), and argues that these are often conflated in prior work. It presents mathematical definitions for both types and categories 415 papers along several dimensions, including the type of bias being addressed, and the evaluation methods used. The authors then explore possible reasons for the inconsistent use of terminology in the literature, and present a case study on both synthetic and real-world datasets illustrating scenarios in which Type I bias occurs in the absence of Type II bias, and vice versa. Finally, it discusses different bias evaluation methods, debiasing methods, and evaluation protocols, addressing each of these specifically for both types of bias.

**Audience:**

Yes

**Broader Impact Concerns:**

I do not have any concerns on the ethical implications of this work.

**Claims And Evidence:**

Yes

**Requested Changes:**

1. Fonts used in tables (e.g. Table 4) and figures (e.g. Figure 8b) are often too small. (Critical)
2. I would suggest to introduce the case studies earlier in the paper as it helps to illustrate the difference between biases. (Optional)

**Strengths And Weaknesses:**

Strengths:
- Discusses an interesting and important issue in the bias literature
- Categorises a large number of bias studies along selected dimension
- Presents case studies demonstrating the difference between the bias types
- Concludes with actionable suggestions for future work

Weaknesses:
- The different types of biases discussed in this paper are already known, partly under different names. For example, these biases have been discussed as "disparate treatment" and "disparate impact" in Barconos and Selbst (2016).
- The future directions subsections are rather short. This work would benefit from a more opinionated discussion that specifically focuses on the differences between the bias types.

[1] Barocas, Solon and Selbst, Andrew D., Big Data's Disparate Impact (2016). 104 California Law Review 671 (2016)

---

> ### Author Response · Authors · 2025-05-24
>
> **W1.**
>
> We appreciate the reviewer’s observation and fully acknowledge that the concepts of disparate treatment and disparate impact, as outlined in Barocas and Selbst (2016) [1], provide foundational legal frameworks for understanding discrimination. However, our work differs from this legal analysis in both scope and the definitions of bias. First, while disparate treatment and disparate impact focus on legal liability in decision-making processes, our paper formalizes Type I Bias and Type II Bias from a machine learning (ML) perspective, introducing mathematical definitions that are directly applicable to evaluating and mitigating algorithmic bias during model development and deployment. Second, the distinction between disparate treatment and disparate impact hinges on whether intent is involved. In contrast, our definitions emphasize performance disparities (Type I Bias) and predictive dependence on protected attributes (Type II Bias), regardless of intent. Notably, the existing ML literature often conflates Type I Bias and Type II Bias, and lacks clear terminology to distinguish between them. Thus, through our taxonomy, unified definitions, and empirical validation, we aim to clarify these distinctions and offer practical guidance for ML researchers—an aspect not addressed in [1].
>
> [1] Barocas, Solon and Selbst, Andrew D., Big Data's Disparate Impact (2016). 104 California Law Review 671 (2016)
>
> **W2.**
>
> We have expanded the future directions subsections to include a more detailed discussion that highlights the distinct challenges and research opportunities associated with each type of bias.
>
>
>
>
> **C1.**
>
> We have enlarged Table 4 as well as Figures 7b and 8b for improved readability.
>
>
> **C2.**
>
> In Section 2 (Definitions), where we mathematically illustrate the distinction between Type I Bias and Type II Bias, we have added indexing to the empirical case study to distinguish between the two bias types.

---

### Author Response · Authors · 2025-05-24

We thank all reviewers for their valuable questions and suggestions.
We appreciate their recognition of our work on an important issue (Reviewer rLRv), the comprehensive nature of our review (Reviewers rLRv, HUqC, B3WE), and the proposed framework and empirical analysis to distinguish between Type I Bias and Type II Bias (Reviewers rLRv, B3WE).
We have revised the paper accordingly and highlighted the modified sections in orange.

---

### Decision · Action_Editor_LGFB · 2025-06-19

**Recommendation:** Accept as is

**Audience:**

Yes

**Audience Explanation:**

It seems it would be helpful for people working in the field to share the vocabulary and differentiate between the two types of biases.

**Claims And Evidence:**

Yes

**Claims Explanation:**

The paper essentially claims that: (a) there is a serious confusion in the field regarding two different types of biases, to which they refer to as the Type I Bias (an uneven performance amoung groups) and Type II Bias (using priviledged attributes); and (b) the two types of bias are fundamentally different in terms of their causes and effects.

There is ample evidence for the confusion in the field. In particular, Authors show that papers tend to use incorrect metrics, or cite papers that discuss one type of bias instead of the other. One convincing example is: "We find that (a) datasets for these tasks contain significant gender bias and (b) models trained on these datasets further amplify existing bias" from one paper, which indeed is not precise enough for scientific writing.

The paper is further supported by experimental results showing the independence between Type I Bias and Type II Bias.

On one hand, it does seem obvious that Type I and Type II Bias are different. On the other hand, indeed there has been this collision in terminology (which perhaps stems from the imprecise way we use the word bias in everyday life?).

All Reviewers recommended acceptance. It is my pleasure to also recommend accepting the work as is. I hope the survey will bring additional clarity to the field. I would love Type I and Type II Bias to become standard vocabulary!